# Mixed Hamiltonian Monte Carlo for Mixed Discrete and Continuous Variables

**Guangyao Zhou**
Vicarious AI
Union City, CA 94587, USA
`stannis@vicarious.com`

## Abstract

Hamiltonian Monte Carlo (HMC) has emerged as a powerful Markov Chain Monte Carlo (MCMC) method to sample from complex continuous distributions. However, a fundamental limitation of HMC is that it can not be applied to distributions with mixed discrete and continuous variables. In this paper, we propose mixed HMC (M-HMC) as a general framework to address this limitation. M-HMC is a novel family of MCMC algorithms that evolves the discrete and continuous variables in tandem, allowing more frequent updates of discrete variables while maintaining HMC's ability to suppress random-walk behavior. We establish M-HMC's theoretical properties, and present an efficient implementation with Laplace momentum that introduces minimal overhead compared to existing HMC methods. The superior performances of M-HMC over existing methods are demonstrated with numerical experiments on Gaussian mixture models (GMMs), variable selection in Bayesian logistic regression (BLR), and correlated topic models (CTMs).

## 1 Introduction

Markov chain Monte Carlo (MCMC) is one of the most powerful methods for sampling from probability distributions. The Metropolis-Hastings (MH) algorithm is a commonly used general-purpose MCMC method, yet is inefficient for complex, high-dimensional distributions because of the random walk nature of its movements. Recently, Hamiltonian Monte Carlo (HMC) [13, 22, 2] has emerged as a powerful alternative to MH for complex continuous distributions due to its ability to follow the curvature of target distributions using gradients information and make distant proposals with high acceptance probabilities. It enjoyed remarkable empirical success, and (along with its popular variant No-U-Turn Sampler (NUTS) [16]) is adopted as the dominant inference strategy in many probabilistic programming systems [8, 27, 3, 25, 14, 10]. However, a fundamental limitation of HMC is that it can not be applied to distributions with mixed discrete and continuous variables.

One existing approach for addressing this limitation involves integrating out the discrete variables(e.g. in Stan[8], Pyro[3]), yet it's only applicable on a small-scale, and can not always be carried out automatically. Another approach involves alternating between updating continuous variables using HMC/NUTS and discrete variables using generic MCMC methods (e.g. in PyMC3[27], Turing.jl[14]). However, to suppress random walk behavior in HMC, long trajectories are needed. As a result, the discrete variables can only be updated infrequently, limiting the efficiency of this approach. The most promising approach involves updating the discrete and continuous variables in tandem. Since naively making MH updates of discrete variables within HMC results in incorrect samples [22], novel variants of HMC (e.g. discontinuous HMC (DHMC)[23, 29], probabilistic path HMC (PPHMC) [12]) are developed. However, these methods can not be easily generalized to complicated discrete state spaces (DHMC works best for ordinal discrete parameters, PPHMC is only applicable to phylogenetic trees), and as we show in Sections 2.5 and 3, DHMC's embedding and algorithmic structure are inefficient.

In this paper, we propose mixed HMC (M-HMC), a novel family of MCMC algorithms that better addresses this limitation. M-HMC provides a general mechanism, applicable to any distributions with mixed support, to evolve the discrete and continuous variables in tandem. It allows more frequent updates of discrete variables while maintaining HMC's ability to suppress random walk behavior, and adopts an efficient implementation (using Laplace momentum) that introduces minimal overhead compared to existing HMC methods. In Section 2, we review HMC and some of its variants involving discrete variables, present M-HMC and rigorously establish its correctness, before presenting its efficient implementation with Laplace momentum and an illustrative application to 1D GMM. We demonstrate M-HMC's superior performances over existing methods with numerical experiments on GMMs, BLR and CTMs in Section 3, before concluding with discussions in Section 4.

## 2 Mixed Hamiltonian Monte Carlo (M-HMC)

Our goal is to sample from a target distribution $\pi(x, q^\mathcal{C}) \propto e^{-U(x,q^\mathcal{C})}$ on $\Omega \times \mathbb{R}^{N_c}$ with mixed discrete variables $x = (x_1, \ldots, x_{N_\mathcal{D}}) \in \Omega$ and continuous variables $q^\mathcal{C} = (q_1^\mathcal{C}, \ldots, q_{N_\mathcal{C}}^\mathcal{C}) \in \mathbb{R}^{N_c}$.

### 2.1 Review of HMC and some variants of HMC that involve discrete variables

For a continuous target distribution $\pi(q^\mathcal{C}) \propto e^{-U(q^\mathcal{C})}$, the original HMC introduces auxiliary momentum variables $p^\mathcal{C} \in \mathbb{R}^{N_c}$ associated with a kinetic energy function $K^\mathcal{C}$, and draws samples for $\pi(q^\mathcal{C})$ by sampling from the joint distribution $\pi(q^\mathcal{C})\chi(p^\mathcal{C})(\chi(p^\mathcal{C}) \propto e^{-K^\mathcal{C}(p^\mathcal{C})})$ with simulations of

$$\frac{\mathrm{d}q^\mathcal{C}(t)}{\mathrm{d}t} = \nabla K^\mathcal{C}(p^\mathcal{C}), \frac{\mathrm{d}p^\mathcal{C}(t)}{\mathrm{d}t} = -\nabla U(q^\mathcal{C}) \text{ (Hamiltonian dynamics)}$$

A foundational tool in applying HMC to distributions with discrete variables is the discontinuous variant of HMC, which operates on piecewise continuous potentials. This was first studied in [24], where the authors proposed binary HMC to sample from binary distributions $\pi(x) \propto e^{-U(x)}$ for $x \in \Omega = \{-1, 1\}^{N_\mathcal{D}}$. The idea is to embed the binary variables $x$ into the continuum by introducing auxiliary location variables $q^\mathcal{D} \in \mathbb{R}^{N_\mathcal{D}}$ associated with a conditional distribution

$$\psi(q^\mathcal{D}|x) : \begin{cases} \psi(q^\mathcal{D}|x) \propto \begin{cases} e^{-\frac{1}{2}\sum_{i=1}^{N_d}(q_i^\mathcal{D})^2} & \text{(Gaussian)} \\ e^{-\sum_{i=1}^{N_\mathcal{D}}|q_i^\mathcal{D}|} & \text{(Exponential)} \end{cases} & \text{If } sign(q_i^\mathcal{D}) = x_i, \forall i = 1, \cdots, N^\mathcal{D} \\ \psi(q^\mathcal{D}|x) = 0 & \text{Otherwise} \end{cases}$$

Binary HMC introduces auxiliary momentum variables $p^\mathcal{D} \in \mathbb{R}^{N_\mathcal{D}}$ associated with a kinetic energy $K^\mathcal{D}(p^\mathcal{D}) = \sum_{i=1}^{N_\mathcal{D}}(p_i^\mathcal{D})^2/2$, and operates on the joint distribution $\Psi(q^\mathcal{D})\nu(p^\mathcal{D})(\nu(p^\mathcal{D}) \propto e^{-K^\mathcal{D}(p^\mathcal{D})})$ on $\Sigma = \mathbb{R}^{N_\mathcal{D}} \times \mathbb{R}^{N_\mathcal{D}}$. The distribution $\Psi(q^\mathcal{D}) = \sum_{x \in \Omega} \pi(x)\psi(q^\mathcal{D}|x)$ gives rise to a piecewise continuous potential, and [24] developed a way to exactly integrate Hamiltonian dynamics for $\Psi(q^\mathcal{D})\nu(p^\mathcal{D})$, taking into account discontinuities in the potential. $x$ and $q^\mathcal{D}$ are coupled through signs of $q^\mathcal{D}$ in $\psi$, so we can read out samples for $x$ from the signs of binary HMC samples for $q^\mathcal{D}$. We show in supplementary that binary HMC is a special case of M-HMC, with Gaussian/exponential binary HMC corresponding to two particular choices of $k^\mathcal{D}$ (defined in Section 2.2) in M-HMC.

[21] later made the key observation that we can analytically integrate Hamiltonian dynamics with piecewise continuous potentials near a discontinuity while perserving the total (potential and kinetic) energy. The trick is to calculate the potential energy difference $\Delta E$ across an encountered discontinuity, and either refract (replace $p_\perp^\mathcal{D}$, the component of $p^\mathcal{D}$ that's perpendicular to the discontinuity boundary, by $\sqrt{\frac{1}{2}||p_\perp^\mathcal{D}||^2 - \Delta E}(p_\perp^\mathcal{D}/||p_\perp^\mathcal{D}||))$ if there's enough kinetic energy ($\frac{1}{2}||p_\perp^\mathcal{D}||^2 > \Delta E$), or reflect (replace $p_\perp^\mathcal{D}$ by $-p_\perp^\mathcal{D}$) if there is not enough kinetic energy ($\frac{1}{2}||p_\perp^\mathcal{D}||^2 \leq \Delta E$). Reflection/refraction HMC (RRHMC) combines the above observation with the leapfrog integrator, and generalizes binary HMC to arbitrary piecewise continuous potentials with discontinuities across affine boundaries. However, RRHMC is computationally expensive due to the need to detect all encountered discontinuities, and by itself can not directly handle distributions with mixed support.

[23] proposed DHMC as an attempt to address some of the issues of RRHMC. It uses Laplace momentum to avoid the need to detect encountered discontinuities, and handles discrete variables (which it assumes take positive integer values, i.e. $x \in \mathbb{Z}_+^{N_\mathcal{D}}$) by an embedding into 1D spaces

$(x_i = n \iff q_i^{\mathcal{D}} \in (a_n, a_{n+1}], 0 = a_1 \leq a_2 \leq \cdots)$ and a coordinate-wise integrator (a special case of M-HMC with Laplace momentum as shown in Section 2.4). In Sections 2.5 and 3, using numerical experiments, we show that DHMC's embedding is inefficient and sensitive to ordering, and it can not easily generalize to more complicated discrete state spaces; furthermore, its need to update all discrete variables at every step makes it computationally expensive for long HMC trajectories.

## 2.2   The general framework of M-HMC

Formally, M-HMC operates on the expanded state space $\Omega \times \Sigma$, where $\Sigma = \mathbb{T}^{N_{\mathcal{D}}} \times \mathbb{R}^{N_{\mathcal{D}}} \times \mathbb{R}^{N_C} \times \mathbb{R}^{N_C}$ with auxiliary location variables $q^{\mathcal{D}} \in \mathbb{T}^{N_{\mathcal{D}}}$ and momentum variables $p^{\mathcal{D}} \in \mathbb{R}^{N_{\mathcal{D}}}$ for $x \in \Omega$, and auxiliary momentum variables $p^{\mathcal{C}} \in \mathbb{R}^{N_C}$ for $q^{\mathcal{C}} \in \mathbb{R}^{N_C}$. Here $\mathbb{T}^{N_{\mathcal{D}}} = \mathbb{R}^{N_{\mathcal{D}}}/\tau \mathbb{Z}^{N_{\mathcal{D}}}$ denotes the $N_{\mathcal{D}}$-dimensional flat torus, and is identified as the hypercube $[0, \tau]^{N_{\mathcal{D}}}$ with the 0's and $\tau$'s in different dimensions glued together. We associate $q^{\mathcal{D}}$ with a flat potential $U^{\mathcal{D}}(q^{\mathcal{D}}) = 0, \forall q^{\mathcal{D}} \in \mathbb{T}^{N_{\mathcal{D}}}$ and $p^{\mathcal{D}}$ with a kinetic energy $K^{\mathcal{D}}(p^{\mathcal{D}}) = \sum_{i=1}^{N_{\mathcal{D}}} k^{\mathcal{D}}(p_i^{\mathcal{D}}), p^{\mathcal{D}} \in \mathbb{R}^{N_{\mathcal{D}}}$ where $k^{\mathcal{D}} : \mathbb{R} \to \mathbb{R}^+$ is some kinetic energy, and $p^{\mathcal{C}}$ with a kinetic energy[1] $K^{\mathcal{C}} : \mathbb{R}^{N_C} \to \mathbb{R}^+$. Use $Q_i, i = 1, \dots, N_{\mathcal{D}}$ to denote $N_{\mathcal{D}}$ irreducible single-site MH proposals, where $Q_i(\tilde{x}|x) > 0$ only when $\tilde{x}_j = x_j, \forall j \neq i$, and $Q_i(x|x) = 0$. In other words, $Q_i$ only changes the value of $x_i$, and always moves away from $x$.

Intuitively, M-HMC also "embeds" the discrete variables $x$ into the continuum (in the form of $q^{\mathcal{D}}$). However, the "embedding" is done by combining the original discrete state space $\Omega$ with the flat torus $\mathbb{T}^{N_{\mathcal{D}}}$: instead of relying on the embedding structure (e.g. the sign of $q_i^{\mathcal{D}}$ in binary HMC, or the value of $q_i^{\mathcal{D}}$ in DHMC) to determine $x$ from $q^{\mathcal{D}}$, in M-HMC we explicitly record the values of $x$ as we can not read out $x$ from $q^{\mathcal{D}}$. $\mathbb{T}^{N_{\mathcal{D}}}$ bridges $x$ with the continuous Hamiltonian dynamics, and functions like a "clock": the system evolves $q_i^{\mathcal{D}}$ with speed determined by the momentum $p_i^{\mathcal{D}}$ and makes an attempt to move to a different state for $x_i$ when $q_i^{\mathcal{D}}$ reaches 0 or $\tau$. Such mixed embedding makes M-HMC easily applicable to arbitrary discrete state spaces, but also prevents the use of methods like RRHMC. For this reason, M-HMC introduces probabilistic proposals $Q_i$'s to move around $\Omega$, and probabilistic reflection/refraction actions to handle discontinuities (which now happen at $q_i^{\mathcal{D}} \in \{0, \tau\}$).

More concretely, M-HMC evolves according to the following dynamics: If $q^{\mathcal{D}} \in (0, \tau)^{N_{\mathcal{D}}}$, $x$ remains unchanged, and $q^{\mathcal{D}}, p^{\mathcal{D}}$ and $q^{\mathcal{C}}, p^{\mathcal{C}}$ follow the Hamiltonian dynamics

$$\text{Discrete} \begin{cases} \frac{dq_i^{\mathcal{D}}(t)}{dt} = (k^{\mathcal{D}})'(p_i^{\mathcal{D}}), i = 1, \dots, N_{\mathcal{D}} \\ \frac{dp^{\mathcal{D}}(t)}{dt} = -\nabla U^{\mathcal{D}}(q^{\mathcal{D}}) = 0 \end{cases} \quad \text{Continuous} \begin{cases} \frac{dq^{\mathcal{C}}(t)}{dt} = \nabla K^{\mathcal{C}}(p^{\mathcal{C}}) \\ \frac{dp^{\mathcal{C}}(t)}{dt} = -\nabla_{q^{\mathcal{C}}} U(x, q^{\mathcal{C}}) \end{cases} \quad (1)$$

If $q^{\mathcal{D}}$ hits either 0 or $\tau$ at site $j$ (i.e. $q_j^{\mathcal{D}} \in \{0, \tau\}$), we propose a new $\tilde{x} \sim Q_j(\cdot|x)$, calculate $\Delta E = \log \frac{\pi(x, q^{\mathcal{C}}) Q_j(\tilde{x}|x)}{\pi(\tilde{x}, q^{\mathcal{C}}) Q_j(x|\tilde{x})}$, and either *refract* if there's enough kinetic energy $(k^{\mathcal{D}}(p_j^{\mathcal{D}}) > \Delta E)$:

$$x \leftarrow \tilde{x}, q_j^{\mathcal{D}} \leftarrow \tau - q_j^{\mathcal{D}}, p_j^{\mathcal{D}} \leftarrow \text{sign}(p_j^{\mathcal{D}})(k^{\mathcal{D}})^{-1}(k^{\mathcal{D}}(p_j^{\mathcal{D}}) - \Delta E)$$

or *reflect* if there is not enough kinetic energy $(k^{\mathcal{D}}(p_j^{\mathcal{D}}) \leq \Delta E)$: $x \leftarrow x, q_j^{\mathcal{D}} \leftarrow q_j^{\mathcal{D}}, p_j^{\mathcal{D}} \leftarrow -p_j^{\mathcal{D}}$.

For the discrete component, because of the flat potential $U^{\mathcal{D}}$, we can exactly integrate the Hamiltonian dynamics with arbitrary $k^{\mathcal{D}}$. For the continuous component, given a discrete state $x$ and some time $t > 0$, use $I(\cdot, \cdot, t|x, U, K^{\mathcal{C}}) : \mathbb{R}^{N_C} \times \mathbb{R}^{N_C} \times \mathbb{R}^+ \to \mathbb{R}^{N_C} \times \mathbb{R}^{N_C}$ to denote a reversible, volume-preserving integrator[2] that's irreducible and aperiodic and approximately evolves the continuous part of the Hamiltonian dynamics in Equation 1 for time $t$. Given the current state $x^{(0)}, q^{\mathcal{C}(0)}$, a full M-HMC iteration first resamples the auxiliary variables

$$q_i^{\mathcal{D}(0)} \sim \text{Uniform}([0, \tau]), p_i^{\mathcal{D}(0)} \sim \nu(p) \propto e^{-k^{\mathcal{D}}(p)} \text{ for } i = 1, \dots, N_{\mathcal{D}}, p^{\mathcal{C}(0)} \sim \chi(p) \propto e^{-K^{\mathcal{C}}(p)}$$

then evolves the discrete variables (using exact integration) and continuous variables (using the integrator $I$) in tandem for a given time $T$, before making a final MH correction like in regular HMC. A detailed description of a full M-HMC iteration is given in Section 1 in the supplementary materials.

Note that if we use conditional distributions for $Q_i$, $\Delta E$ would always be 0, the discrete dynamics in Equation 1 plays no role, and M-HMC reduces to the incorrect case of naively making Gibbs updates within HMC. However, the requirement $Q_i(x|x) = 0$ (which is more efficient [19]) means $Q_i$ is always sufficiently different from the conditional distribution and guarantees correctness of M-HMC.

## 2.3 M-HMC samples from the correct distribution

For notational simplicity, define $\Theta = (q^{\mathcal{D}}, p^{\mathcal{D}}, q^{\mathcal{C}}, p^{\mathcal{C}})$. To prove M-HMC samples from the correct distribution $\pi(x, q^{\mathcal{C}})$, we show that a full M-HMC iteration preserves the joint invariant distribution $\varphi((x, \Theta)) \propto \pi(x, q^{\mathcal{C}}) e^{-\left[U^{\mathcal{D}}(q^{\mathcal{D}}) + K^{\mathcal{D}}(p^{\mathcal{D}}) + K^{\mathcal{C}}(p^{\mathcal{C}})\right]}$ and establish its irreducibility and aperiodicity. At each iteration, the resampling can be seen as a Gibbs step, where we resample the auxiliary variables $q^{\mathcal{D}}, p^{\mathcal{D}}, q^{\mathcal{C}}$ from their conditional distribution given $x, q^{\mathcal{C}}$. This obviously preserves $\varphi$. So we only need to prove detailed balance of the evolution of $x$ and $q^{\mathcal{C}}$ in an M-HMC iteration (described in detail in the *M-HMC* function in Section 1 of the supplementary materials) w.r.t. $\varphi$. Formally, $\forall T > 0$, the *M-HMC* function (section 1 of supplementary) defines a transition probability kernel $R_T((x, \Theta), B) = \mathbb{P}(\textit{M-HMC}(x, \Theta, T) \in B), \forall (x, \Theta) \in \Omega \times \Sigma$ and $B \subset \Omega \times \Sigma$ measurable. For all $A \subset \Omega \times \Sigma$ measurable, $\Theta \in \Sigma$, define $A(\Theta) = \{x \in \Omega : (x, \Theta) \in A\}$. We have

**Theorem 1.** *(Detailed Balance) The M-HMC function (Section 1 of supplementary) satisfies detailed balance w.r.t. the joint invariant distribution $\varphi$, i.e. for any measurable sets $A, B \subset \Omega \times \Sigma$,*

$$\int_{\Sigma} \sum_{x \in A(\Theta)} R_T((x, \Theta), B) \varphi((x, \Theta)) \mathrm{d}\Theta = \int_{\Sigma} \sum_{x \in B(\Theta)} R_T((x, \Theta), A) \varphi((x, \Theta)) \mathrm{d}\Theta$$

*Proof Sketch.* Use $s = (x, q^{\mathcal{D}}, p^{\mathcal{D}}, q^{\mathcal{C}}, p^{\mathcal{C}}), s' = (x', q^{\mathcal{D}\prime}, p^{\mathcal{D}\prime}, q^{\mathcal{C}\prime}, p^{\mathcal{C}\prime}) \in \Omega \times \Sigma$ to denote 2 points.

**Sequence of proposals and probabilistic paths**    Starting from $s \in \Omega \times \Sigma$, for a given travel time $T$, a concrete M-HMC iteration involves a finite sequence of realized discrete proposals $Y$. If we fix $Y$, the M-HMC iteration (without the final MH correction) specifies a deterministic mapping from $s$ to some $s'$. For a given $Y$, we introduce an associated probabilistic path $\omega(s, T, Y)$ (containing information on $Y$, indices/times and accept/reject decisions for discrete updates, and evolution of $s$) to describe the deterministic trajectory going from $s$ to $s'$ in time $T$ through the M-HMC iteration.

**Countable number of probabilistic paths and decomposition of $R_T(s, B)$**    Since $T$ and $\Omega$ are finite, traveling from $s$ for time $T$ gives a countable number of possible destinations $s'$. This implies there can only be a countable number of valid probabilistic paths, and we can decompose $R_T(s, B) = \sum_{s'} \sum_Y r_{T,Y}(s, s')$. Here we sum over all possible destinations $s'$ and all valid $Y$'s for which $\omega(s, T, Y)$ brings $s$ to $s'$. $r_{T,Y}(s, s')$ denotes the transition probability along $\omega(s, T, Y)$.

**Proof of detailed balance**    Using similar proof techniques as in RRHMC, we can prove detailed balance for $r_{T,Y}$ (Lemma 4 in supplementary). This in turn proves detailed balance of M-HMC. □

We defer detailed definitions and proofs to the supplementary. Combining the above theorem with irreducibility and aperiodicity (which follow from irreducibility and aperiodicity of the integrator $I$, and the irreducibility of the $Q_i$'s) proves that M-HMC samples from the correct distribution $\pi(x, q^{\mathcal{C}})$.

## 2.4 Efficient M-HMC implementation with Laplace momentum

We next present an efficient implementation of M-HMC using Laplace momentum $k^{\mathcal{D}}(p) = |p|$. While M-HMC works with any $k^{\mathcal{D}}$, using a general $k^{\mathcal{D}}$ requires detection of all encountered discontinuities, similar to RRHMC. However, with Laplace momentum, $q_i^{\mathcal{D}}$'s speed (given by $(k^{\mathcal{D}})'(p_i^{\mathcal{D}})$) becomes a constant 1, and we can precompute the occurences of all discontinuities at the beginning of each M-HMC iteration. In particular, we no longer need to explicitly record $q^{\mathcal{D}}, p^{\mathcal{D}}$, but can instead keep track of only the kinetic energies associated with $x$. Note that we need to use $\tau$ to orchestrate discrete and continuous updates. Here, instead of explicitly setting $\tau$, we propose to alternate discrete and continuous updates, specifying the total travel time $T$, the number of discrete updates $L$, and the number of discrete variables to update each time $n_{\mathcal{D}}$. The step sizes are properly scaled (effectively setting $\tau$) to match the desired total travel time $T$. To reduce integration error and ensure a high acceptance rate, we specify a maximum step size $\varepsilon$. A detailed description of the efficient implementation is given in Algorithm 1. See Section 2 of supplementary for a detailed discussion on how each part of Algorithm 1 can be derived from the original *M-HMC* function in Section 1 of supplementary. The coordinate-wise integrator in DHMC corresponds to setting $n_{\mathcal{D}} = N_{\mathcal{D}}$ with $Q_i$'s that are implicitly specified through embedding. However, the need to update all discrete variables at each step is computationally expensive for long HMC trajectories. In contrast, M-HMC can flexibly orchestrate discrete and continuous updates depending on models at hand, and introduces minimal overhead ($x$ updates that are usually cheap) compared to existing HMC methods.

---
**Algorithm 1** M-HMC with Laplace momentum
---
**Require:** $U$, target potential; $Q_i, i = 1, \ldots, N_{\mathcal{D}}$, single-site proposals; $\varepsilon$, maximum step size; $L$, # of times to update discrete variables; $n_{\mathcal{D}}$, # of discrete sites to update each time

**input** $x^{(0)}$, current discrete state; $q^{\mathcal{C}(0)}$, current continuous location; $T$, travel time

**output** $x$, next discrete state; $q^{\mathcal{C}}$, next continuous location

1: **function** M-HMCLaplaceMomentum($x^{(0)}, q^{\mathcal{C}(0)}, T | U, Q_i, i = 1, \ldots, N_{\mathcal{D}}, \varepsilon, L, n^{\mathcal{D}}$)
2: $\quad k_i^{\mathcal{D}(0)} \sim \text{Exponential}(1), i = 1, \ldots, N_{\mathcal{D}}, p_i^{\mathcal{C}(0)} \sim N(0,1), i = 1, \ldots, N_{\mathcal{C}}$
3: $\quad x \leftarrow x^{(0)}, k^{\mathcal{D}} \leftarrow k^{\mathcal{D}(0)}, q^{\mathcal{C}} \leftarrow q^{\mathcal{C}(0)}, p^{\mathcal{C}} \leftarrow p^{\mathcal{C}(0)}$
4: $\quad \Lambda \sim \text{RandomPermutation}(\{1, \ldots, N_{\mathcal{D}}\})$
5: $\quad (\eta, M) \leftarrow \textit{GetStepSizesNSteps}(\varepsilon, T, L, N_{\mathcal{D}}, n_{\mathcal{D}})$ # Defined in Section 2 of supplementary
6: $\quad$ **for** $t$ **from** 1 **to** $L$ **do**
7: $\quad\quad$ **for** $s$ **from** 1 **to** $M_t$ **do** $q^{\mathcal{C}}, p^{\mathcal{C}} \leftarrow \textit{leapfrog}(q^{\mathcal{C}}, p^{\mathcal{C}}, \eta_t)$ **end for**
8: $\quad\quad$ **for** $s$ **from** 1 **to** $n_{\mathcal{D}}$ **do** $x, k^{\mathcal{D}} \leftarrow \textit{DiscreteStep}(x, k^{\mathcal{D}}, q^{\mathcal{C}}, \Lambda_{[(t-1)n_{\mathcal{D}}+s] \bmod N_{\mathcal{D}}})$ **end for**
9: $\quad$ **end for**
10: $\quad E \leftarrow U\left(x, q^{\mathcal{C}}\right) + \sum_{i=1}^{N_{\mathcal{D}}} k_i^{\mathcal{D}} + K^{\mathcal{C}}(p^{\mathcal{C}}), E^{(0)} \leftarrow U\left(x^{(0)}, q^{\mathcal{C}(0)}\right) + \sum_{i=1}^{N_{\mathcal{D}}} k_i^{\mathcal{D}(0)} + K^{\mathcal{C}}(p^{\mathcal{C}(0)})$
11: $\quad$ **if** $\text{Uniform}([0,1]) >= e^{-(E-E^{(0)})}$ **then** $x \leftarrow x^{(0)}, q^{\mathcal{C}} \leftarrow q^{\mathcal{C}(0)}$ **end if**
12: $\quad$ **return** $x, q^{\mathcal{C}}$
13: **end function**
---
14: **function** leapfrog($q^{\mathcal{C}}, p^{\mathcal{C}}, \tilde{\varepsilon}$)
15: $\quad p^{\mathcal{C}} \leftarrow p^{\mathcal{C}} - \tilde{\varepsilon}\nabla_{q^{\mathcal{C}}} U(x, q^{\mathcal{C}})/2; q^{\mathcal{C}} \leftarrow q^{\mathcal{C}} + \tilde{\varepsilon}p^{\mathcal{C}}; p^{\mathcal{C}} \leftarrow p^{\mathcal{C}} - \tilde{\varepsilon}\nabla_{q^{\mathcal{C}}} U(x, q^{\mathcal{C}})/2$
16: $\quad$ **return** $q^{\mathcal{C}}, p^{\mathcal{C}}$
17: **end function**
---
18: **function** DiscreteStep($x, k^{\mathcal{D}}, q^{\mathcal{C}}, j$)
19: $\quad \tilde{x} \sim Q_j(\cdot|x); \Delta E \leftarrow \log \frac{e^{-U(x,q^{\mathcal{C}})}Q_j(\tilde{x}|x)}{e^{-U(\tilde{x},q^{\mathcal{C}})}Q_j(x|\tilde{x})}$
20: $\quad$ **if** $k_j^{\mathcal{D}} > \Delta E$ **then** $x \leftarrow \tilde{x}, k_j^{\mathcal{D}} \leftarrow k_j^{\mathcal{D}} - \Delta E$ **end if**
21: $\quad$ **return** $x, k^{\mathcal{D}}$
22: **end function**
---

## 2.5 Illustrative application of M-HMC to 1D Gaussian mixture model (GMM)

In this section, we illustrate some important aspects of M-HMC by applying M-HMC to a concrete 1D GMM with 4 mixture componets. Use $x \in \{1, 2, 3, 4\}$ to denote the discrete variable, and $q^{\mathcal{C}} \in \mathbb{R}$ to denote the continuous variable. We study the 1D GMM $\pi(x, q^{\mathcal{C}}) = \phi_x N(q^{\mathcal{C}}|\mu_x, \Sigma)$, where $\phi_1 = 0.15, \phi_2 = \phi_3 = 0.3, \phi_4 = 0.25, \Sigma = 0.1$, and $\mu_1 = -2, \mu_2 = 0, \mu_3 = 2, \mu_4 = 4$.

**More frequent discrete updates within HMC are beneficial** The essential idea of M-HMC is to evolve discrete and continuous variables in tandem, allowing more frequent discrete updates within HMC. Figure 2(a) visualizes the evolution of $x, q^{\mathcal{C}}$ in an M-HMC iteration on our 1D GMM, and intuitively shows the benefits of such more frequent discrete updates: M-HMC can make frequent attempts to move to a different mixture component; such attempts can often succeed when M-HMC gets close to a different mixture component while traversing the current one; the ability to move to different mixture components within an M-HMC iteration allows M-HMC to make distant proposals, which are accepted with high probabilities due to the use of HMC-like mechanisms. Figure 2(a) demonstrates one such distant proposal in which M-HMC moves across all 4 mixture components in one iteration. Such distant proposals are unlikely to happen in methods that alternate between HMC and discrete updates, limiting the efficiency of such methods. In Section 3, we would further demonstrate the efficiency of M-HMC when compared with alternatives using numerical experiments.

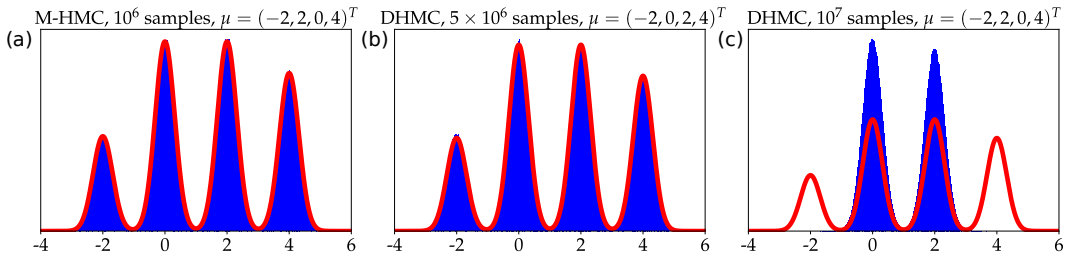

Figure 1: Samples histograms (blue) and true density (red) on 1D GMM for M-HMC and DHMC

**Naively making discrete updates within HMC is incorrect**   Figure 2(left) compares naive MH within HMC (MHwHMC) and M-HMC for 1D GMM. The seemingly trivial distinction naturally comes out of Algorithm 1 with 1 discrete variable, yet corrects the inherent bias in MHwHMC (see Figure 2(b)(c)). This demonstrates the necessity to use the M-HMC framework to evolve discrete and continuous variables in tandem. See Section 3 of supplementary materials for more details.

**M-HMC is applicable to arbitrary distributions with mixed support, unlike DHMC**   DHMC does not easily generalize to complicated discrete state spaces due to its 1D embedding. A simple illustration is to apply DHMC to 1D GMM, but instead with $\mu_2 = 2, \mu_3 = 0$. While the model remains exactly the same, as shown in red curves in Figures 1(b)(c), due to its sensitivity to the ordering of discrete states, DHMC failed to sample all components even after $10^7$ samples (Figure 1(c)), even though it can fit well with $5 \times 10^6$ samples in the original setup (Figure 1(b)). In contrast, M-HMC suffers no such issue, and works well in both cases with $10^6$ samples (Figures 1(a) and 2(c)), and in general for arbitrary distributions with mixed support. See Section 3.3 for another example.

## 3   Numerical experiments

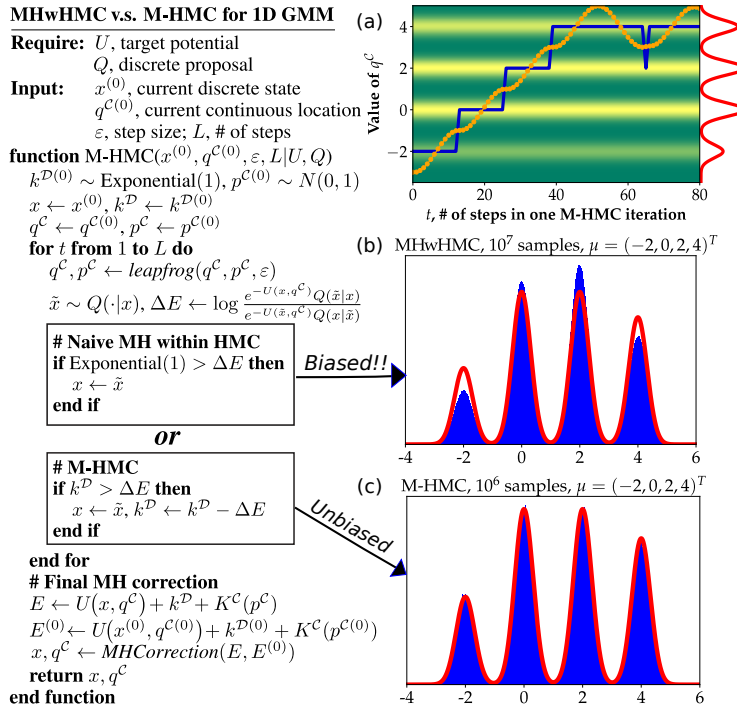

Figure 2: Proposed M-HMC kernel and comparison of MHwHMC and M-HMC on 1D GMM. **Figure 2(a):** Evolution of $x$ (in the form of $\mu_x$, blue) and $q^\mathcal{C}$ (orange) in an M-HMC iteration. Background color and red curve visualize model density. **Figure 2(left):** Comparison of MHwHMC and M-HMC on 1D GMM. **Figure 2(b)(c):** Samples histograms (blue) and true density (red) for MHwHMC and M-HMC.

In this section, we empirically verify the accuracy of M-HMC, and compare the performances of various samplers for GMMs, variable selection in BLR, and CTM. In addition to DHMC and M-HMC, we also compare NUTS (using Numpyro [25], for GMMs), HMC-within-Gibbs (HwG), NUTS-within-Gibbs (NwG, implemented as a compound step in PyMC3 [27]), and specialized Gibbs samplers (adapting [26] for variable selection in BLR, and adapting [9] for CTM). Our implementations of DHMC, M-HMC and HwG rely on *JAX* [6].   For Gibbs samplers, we combine *NUMBA* [28] with the package *pypolyagamma*[3]. The exact parameter values for different samplers can be found in the supplementary, and in the code to reproduce the results[4].

For all three models, a common performance measure is the minimum relative effective sample size (MRESS), i.e. the minimum ESS over all dimensions, normalized by the number of samples. We use function *ess* (with default settings) from Python package *arviz* [18] to estimate MRESS. Our MRESS is estimated using multiple independent chains. For discrete updates in HwG and NwG, in addition to the MH updates used in our experiments, we also tried standard particle Gibbs (using *Turing.jl* [14]) as suggested by an anonymous reviewer, but were unable to get meaningful results due to numerical accuracy in *Turing.jl* implementations. For experiments with M-HMC, we use $Q_j(\tilde{x}|x) \propto \pi(\tilde{x})\rho_j(\tilde{x}|x)$, where $\rho_j(\tilde{x}|x) = \begin{cases} 1 & \text{if } \tilde{x}_j \neq x_j, \tilde{x}_i = x_i, i \neq j \\ 0 & \text{otherwise} \end{cases}$ [19], as required in Section 2.2. Such efficient [19] proposals are also used in other samplers to ensure fair comparison.

[3]For efficient sampling from Polya-Gamma distribution. `github.com/slinderman/pypolyagamma`
[4]Code available at `https://github.com/StannisZhou/mixed_hmc`

## 3.1 24D Gaussian Mixture Model (GMM)

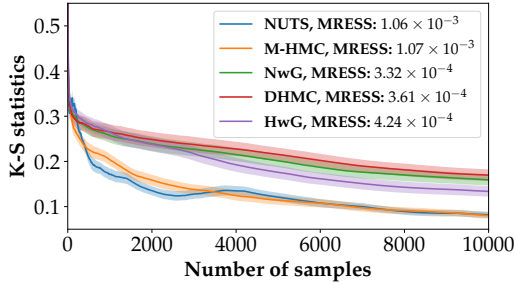

Figure 3: Evolution of K-S statistics of empirical and true samples for $q_1^{\mathcal{C}}$, and MRESS for the 24D GMM. Colored regions indicate 95% confidence interval, estimated using 192 independent chains.

We experiment with a more challenging 24D GMM with 4 components. We again use $\phi_1 = 0.15, \phi_2 = \phi_3 = 0.3, \phi_4 = 0.25$. To avoid potential intractability because of multimodality, we set $\Sigma = 3I$. We use the 24 permutations of $-2, 0, 2, 4$ to specify the means of the 4 components in the 24 dimensions. We test 5 different samplers: NUTS, HwG, NwG, DHMC and M-HMC. NUTS operates on the marginal distribution $\pi(q^{\mathcal{C}})$, and serves to provide an upper bound on the performance. All other samplers operate on the joint distribution $\pi(x, q^{\mathcal{C}})$.

NUTS and NwG require no tuning. We favor HwG and DHMC with a parameter grid search, and tune M-HMC by inspecting short trial runs. For each sampler, we draw $10^4$ burn-in and $10^4$ actual samples in 192 independent chains.

To get a sense of the accuracy of the samplers as well as their convergence speed, we calculate the two-sided Kolmogorov-Smirnov (K-S) statistic[5] of the 24 marginal empirical distributions given by samples from the samplers and the true marginal distributions, averaged over 192 chains. We also calculate the MRESS for $q^{\mathcal{C}}$ to measure the efficiency of the different samplers. Figure 3 shows the evolution of the K-S statistic for $q_1^{\mathcal{C}}$, with MRESS reported in legends. M-HMC clearly outperforms HwG, NwG and DHMC, and is even on par with NUTS[6], which explicitly integrates out $x$. DHMC and NwG have essentially the same performance, and are slightly outperformed by HwG.

## 3.2 Variable Selection in Bayesian Logistic Regression (BLR)

We consider the logistic regression model $y_i \sim \text{Bernoulli}\left(\sigma(X_i^T \beta)\right), i = 1, \cdots, 100$ where $X \in \mathbb{R}^{100 \times 20}, \beta \in \mathbb{R}^{20}$, and $\sigma(x) = 1/(1 + e^{-x})$ is the sigmoid function. For our experiments, we generate a set of synthetic data: The $X_i$'s are generated from the multivariate Gaussian $N(0, \Sigma)$, where $\Sigma_{jj} = 3, j = 1, \cdots, 20$ and $\Sigma_{jk} = 0.3, \forall j \neq k$. For $\beta$, we set 5 randomly picked components to be 0.5, and all the other components to be 0. We generate $y_i \sim \text{Bernoulli}\left(\sigma(X_i^T \beta)\right)$. We introduce a set of binary random variables $\gamma_j, j = 1, \cdots, 20$ to indicate the presence of components of $\beta$, and put an uninformative prior $N(0, 25I)$ on $\beta$. This results in the following joint distribution on $\beta, \gamma$ and $y$: $p(\beta, \gamma, y) = N(\beta | 0, 25I) \prod_{i=1}^{100} p_i^{y_i}(1 - p_i)^{1-y_i}$ where $p_i = \sigma(\sum_{j=1}^{20} X_{ij} \beta_j \gamma_j), i = 1, \cdots, 100$.

We are interested in a sampling-based approach to identify the relevant components of $\beta$. A natural approach [11, 30] is to sample from the posterior distribution $p(\beta, \gamma | y)$, and inspect the posterior samples of $\gamma$. This constitutes a challenging posterior sampling problem due to the lack of conjugacy and the mixed support, and prevents the wide applicability of this approach. Existing methods typically rely on data-augmentation schemes [1, 7, 17, 26]. Here we explore applications of HwG, NwG, DHMC and M-HMC to this problem. As a baseline, we implement a specialized Gibbs sampler, by combining the Gibbs sampler in [26] for $\beta$ with a single-site systematic scan Gibbs sampler for $\gamma$.

Gibbs and NwG require no tuning. For HwG and DHMC, we conduct a parameter grid search, and report its best performance. For M-HMC, instead of picking a particular setting, we test its performance on multiple settings, to better understand how different components of M-HMC affect its performance. In particular, we are interested in how performance changes with the number of discrete updates $L$ for a fixed travel time $T$, and with $n_{\mathcal{D}}$, the number of discrete variables to update at each discrete update while holding the total numer of single discrete variable updates $n_{\mathcal{D}}L$ a constant. For each sampler, we use 192 independent chains, each with 1000 burn-in and 2000 actual samples.

We check the accuracy of the samplers by looking at their accuracy in terms of percentage of the posterior samples for $\gamma$ that agree exactly with the true model, as well as their average Hamming

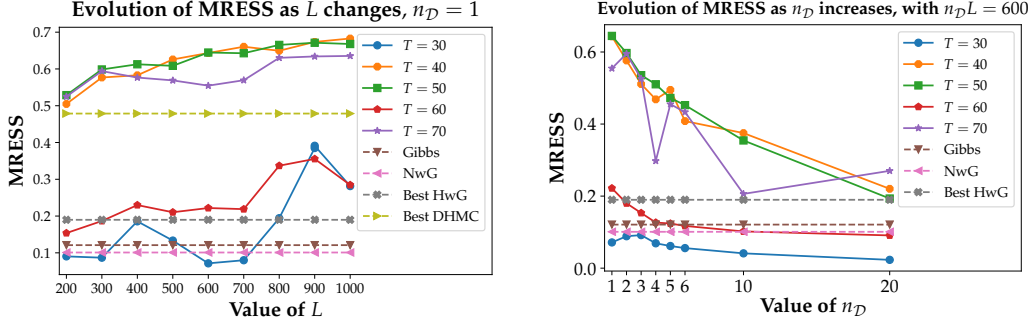

(a) Baseline MRESS for the Gibbs sampler, NwG, and best DHMC, and evolution of MRESS for M-HMC as $L$ changes for different travel time $T$, with $n_{\mathcal{D}} = 1$

(b) Baseline MRESS for the Gibbs sampler, NwG, and evolution of MRESS for M-HMC as $n_{\mathcal{D}}$ increases for different travel time $T$, with $n_{\mathcal{D}}L = 600$

Figure 4: Performances (MRESS of posterior samples for $\beta$) of M-HMC as $L$ and $n_{\mathcal{D}}$ change on variable selection for BLR, as well as baseline MRESS for the Gibbs sampler, NwG, and best DHMC

distance to the true model. All the tested samplers perform similarly, giving about $8.1\%$ accuracy and an average Hamming distance of around $2.2$. We compare the efficiency of the 5 samplers by measuring MRESS of posterior samples for $\beta$. The results are summarized in Figures 4(a)(b). M-HMC and DHMC both significantly outperform Gibbs, HwG and NwG, demonstrating the benefits of more frequent discrete updates inside HMC. However, we observe a "U-turn" [16] phenomenon, shown in Figure 4a, for both $T$ and $L$: increasing $T, L$ results in performance oscillations, suggesting that although M-HMC is capable of making distant proposals, increasing $T, L$ beyond a certain threshold would decrease its efficiency as M-HMC starts to "double back" on itself. Nevertheless, it's clear that for fixed $T$, increasing $L$ generally improves performance, again demonstrating the benefits of more frequent discrete variables updates. We also observe (Figure 4(b)) that $n_{\mathcal{D}} = 1$ generally gives the best performance when $n_{\mathcal{D}}L$ is held as a constant, suggesting that distributed/more frequent updates of the discrete variables is more beneficial than concentrated/less frequent updates. However, distributed/more frequent updates of discrete variables entail using a large $L$, which can break each leapfrog step into smaller steps, resulting in more (potentially expensive) gradients evaluations.

Although the best DHMC has good performance, we note that its algorithmic structure requires sequential updates of all discrete variables at each leapfrog step. Compared with, e.g. M-HMC with $T = 40, L = 600, n_{\mathcal{D}} = 1$, using similar implementations, the best DHMC takes 2.23 times longer with nearly 0.2 reduction in MRESS, demonstrating the superior performance of M-HMC.

### 3.3 Correlated Topic Model (CTM)

Topic modeling is widely used in the statistical analysis of documents collections. CTM [4] is a topic model that extends the popular Latent Dirichlet Allocation (LDA) [5] by using a logistic-normal prior to effectively model correlations among different topics. Our setup follows [4]: assume we have a CTM modeling $D$ documents with $K$ topics and a $V$-word vocabulary. The $K$ topics are specified by a $K \times V$ matrix $\beta$. The $k$th row $\beta_k$ is a point on the $V - 1$ simplex, defining a distribution on the vocabulary. Use $w_{d,n} \in \{1, \cdots, V\}$ to denote the $n$th word in the $d$th document, $z_{d,n} \in \{1, \cdots, K\}$ to denote the topic assignment associated with the word $w_{d,n}$, and use $\mathrm{Categ}(p)$ to denote a categorical distribution with distribution $p$. Define $f : \mathbb{R}^K \to \mathbb{R}^K$ to be $f_i(\eta) = e^{\eta_i} / \sum_{j=1}^{K} e^{\eta_k}$. Given the topics $\beta$, a vector $\mu \in \mathbb{R}^K$ and a $K \times K$ covariance matrix $\Sigma$, for the $d$th document with $N_d$ words, CTM first samples $\eta_d \sim N(\mu, \Sigma)$; then for each $n \in \{1, \cdots, N_d\}$, CTM draws topic assignment $z_{d,n}|\eta_d \sim \mathrm{Categ}(f(\eta_d))$, before finally drawing word $w_{d,n}|z_{d,n}, \beta \sim \mathrm{Categ}(\beta_{z_{d,n}})$.

While CTM has proved to be a better topic model than LDA [4], its use of the non-conjugate logistic-normal prior makes efficient posterior inference of $p(\eta, z|w; \beta, \mu, \Sigma)$ highly challenging. In [4], the authors resorted to a variational inference method with highly idealized mean-field approximations. There has been efforts on developing more efficient inference methods using a sampling-based approach, e.g. specialized Gibbs samplers [20, 9]. In this section, we explore the applications of HwG, NwG, DHMC and M-HMC to the posterior inference problem $p(\eta, z|w; \beta, \mu, \Sigma)$ in CTM.

We use the Associated Press (AP) dataset [15][7], which consists of 2246 documents. Since we are interested in comparing the performance of different samplers, we train a CTM using *ctm-c*[8], with the default settings, $K = 10$ topics and the given vocabulary of $V = 10473$ words. As a baseline, we use the Gibbs sampler developed in [9], which was empirically demonstrated to be highly effective. Note that unlike [9], there's no Dirichlet prior on $\beta$ in our setup; moreover, for $K$ topics, *ctm-c* handles the issue of non-identifiability by using $\eta_d \in \mathbb{R}^{K-1}$ and assuming the first dimension to be 0. Nevertheless, it's straightforward to adapt [9] to our setup. After training with *ctm-c*, we apply the 4 different samplers to 20 randomly picked documents for posterior sampling of $z$ and $\eta$. For each sampler, we draw 1000 burn-in and 4000 actual samples in each of 96 independent chains. Gibbs and NwG require no tuning. For HwG and DHMC, we conduct a parameter grid search. For M-HMC, we inspect short trial runs on a separate document, and fix $T, n_{\mathcal{D}}$ for all 20 picked documents and set $L = 80 \times N_d$ for document $d$. Empirically, we find it important to use a non-identity mass matrix for the kinetic energy $K^{\mathcal{C}}$ in M-HMC, which we implement by using step size $\frac{4\Sigma_{ii}}{\sum_{j=1}^{9} \Sigma_{jj}}$ for $\eta_{d,i}$.

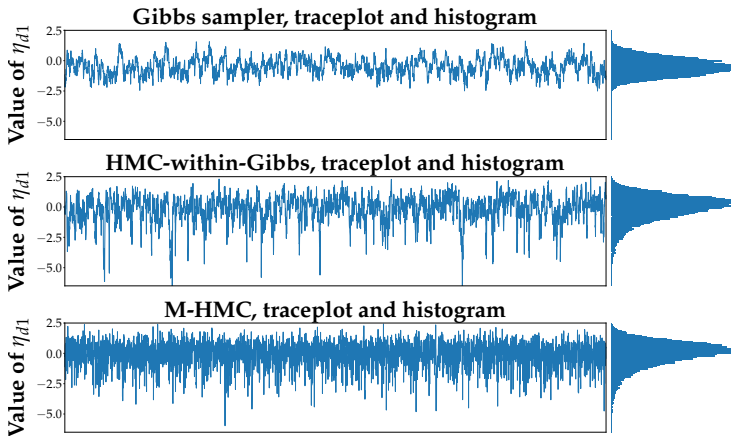

Figure 5: Traceplots and samples histograms of posterior samples of $\eta_{d1}$ when Gibbs differs from HwG, NwG&M-HMC in posterior means

We first compare the accuracy of the 5 different samplers, by inspecting the posterior means of $\eta_d$ using samples from the 5 different samplers on the 20 randomly picked documents. Likely due to its inability to generalize to complicated discrete state spaces, the sample means for $\eta_d$ from DHMC differ significantly from the 4 other samplers on all 20 documents. HwG, NwG and M-HMC agree on all 20 documents, while Gibbs agrees ($\pm 5\%$ relative error) with them on 17 out of the 20 documents.

On the 17 documents where the 4 samplers agree, we calculate MRESS for $\eta_d$. Without much tunning, M-HMC already shows significant advantages: it has the largest MRESS for all 17 documents, and its MRESS is on average **22.48** times larger than that of Gibbs, **3.38** times larger than that of NwG, and **3.3** times larger than that of HwG. HwG slightly outperforms NwG, with Gibbs performing the worst. Note that Gibbs sequentially updates each component of $z$ and $\eta$, which likely causes slow mixing.

We additionally inspect traceplots and samples histograms of posterior samples for $\eta_{d1}$ on a document where Gibbs disagrees with the other 3 samplers (Figure 5. NwG is excluded since it behaves similarly to HwG but is less efficient). M-HMC clearly mixes the fastest, with HwG also outperforming Gibbs. Moreover, HwG and M-HMC explore the state space much more thoroughly, suggesting that Gibbs gives different posterior means on the 3 documents due to ineffective exploration of the state spaces.

## 4   Discussions and Conclusions

Numerical experiments in Sections 2.5 and 3 show that:(1) M-HMC gives accurate samples on all the tested models, while some alternatives occasionally fail (e.g. DHMC in Section 2.5, and Gibbs and DHMC in Section 3.3). (2) In terms of MRESS, M-HMC is consistently more efficient than HwG, NwG, DHMC and Gibbs, and even matches NUTS for 24D GMM. (3) As shown in Section 3.2, M-HMC's performance is sensitive to parameter choices, similar to regular HMC. This makes automatically picking the parameters (e.g. in a NUTS-like way) an important future direction.

Overall, M-HMC provides a generally applicable mechanism that can be easily implemented to make more frequent updates of discrete variables within HMC. Such updates are usually inexpensive (when compared to gradients evaluations) yet highly beneficial as shown in our numerical experiments in Section 3. This makes M-HMC an appealing option for probabilistic models with mixed support.

## Broader Impact

Probabilistic modeling with structured models leads to more interpretable modeling of data and proper uncertainty quantification. M-HMC enables efficient inference for probabilistic models with mixed support, allowing applicability of probabilistic modeling to a broader set of problems. This can contribute to more principled and interpretable decision making process based on probabilistic modeling of data. As with any technology, negative consequences are possible but difficult to predict at this time. This is not a deployed system with immediate failure consequences or that can leverage potentially harmful biases.

## Acknowledgments and Disclosure of Funding

The author would like to thank Stuart Geman for providing the initial spark for this work and many helpful discussions, Nishad Gothoskar for suggesting the name M-HMC, and Rajeev Rikhye for valuable help in improving the figures and poster for the paper. This work was partially supported by the National Science Foundation under Grant No. DMS-1439786 while the author was in residence at the Institute for Computational and Experimental Research in Mathematics in Providence, RI, during the Spring 2019 semester, and by Vicarious AI.

## Footnotes

[1]The simplest choice for $K^{\mathcal{C}}$ is $K^{\mathcal{C}}(p^{\mathcal{C}}) = \sum_{i=1}^{N_C} \frac{(p_i^{\mathcal{C}})^2}{2}$, but M-HMC can work with any kinetic energy.

[2]An example is the commonly used leapfrog integrator

[5]Calculated using *scipy.stats.ks_2samp*

[6]The NUTS adaption is done via dual averaging, with 0.6 target acceptance probability. Note that if we use the default 0.8 in *NumPyro*, NUTS's MRESS reduces to $8.27 \times 10^{-4}$, and is worse than that of M-HMC.

[7]The dataset can be downloaded at `http://www.cs.columbia.edu/~blei/lda-c/ap.tgz`

[8]`https://github.com/blei-lab/ctm-c`

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
