[Supplementary Material · supplementary.pdf]

# Supplement for "Mixed Hamiltonian Monte Carlo for Mixed Discrete and Continuous Variables"

**Guangyao Zhou**
Vicarious AI
Union City, CA 94587, USA
`stannis@vicarious.com`

## 1 Algorithm and theory

### 1.1 Detailed description of a full M-HMC iteration

See Algorithm 1 for a detailed description of a full M-HMC iteration.

### 1.2 Proof of Theorem 1

#### 1.2.1 Proof of the Theorem

**Theorem 1.** *(Detailed Balance) The M-HMC function in Algorithm 1 satisfies detailed balance w.r.t. the joint invariant distribution $\varphi$, i.e. for any measurable sets $A, B \subset \Omega \times \Sigma$,*

$$\int_{\Sigma} \sum_{x \in A(\Theta)} R_T((x,\Theta), B)\varphi((x,\Theta))\mathrm{d}\Theta = \int_{\Sigma} \sum_{x \in B(\Theta)} R_T((x,\Theta), A)\varphi((x,\Theta))\mathrm{d}\Theta$$

*Proof.* Use $s = (x, q^{\mathcal{D}}, p^{\mathcal{D}}, q^{\mathcal{C}}, p^{\mathcal{C}})$ and $s' = (x', q^{\mathcal{D}\prime}, p^{\mathcal{D}\prime}, q^{\mathcal{C}\prime}, p^{\mathcal{C}\prime})$ to denote two points in $\Omega \times \Sigma$.

**Sequence of proposals and probabilistic paths**

If we start from $s \in \Omega \times \Sigma$, for a given travel time $T$, a concrete run of the *M-HMC* function would involve a finite sequence of random proposals. Assume the length of the sequence is $M$. The sequence of random proposals $Y$ can be denoted as

$$Y = (y^{(0)}, y^{(1)}, \ldots, y^{(M-1)}), y^{(m)} \in \Omega, m = 0, \ldots, M-1$$

This sequence of proposals indicates that, for this particular run of *M-HMC*, we reach 0 or $\tau$ at individual sites $M$ times, and each time the system makes a proposal to go to the discrete state $y^{(m)} \in \Omega, m = 0, \cdots, M-1$ from the current discrete state.

If we fix $Y$, the *M-HMC* function (without the final accept/reject step) in fact specifies a deterministic mapping, and would map $s$ to a single point $s' \in \Omega \times \Sigma$. For each such sequence of proposals $Y$, we introduce an associated probabilistic path $\omega(s, T, Y)$, which contains all the information of the system going from $s$ to $s'$ in time $T$ through the function *M-HMC*. Formally, $\omega(s, T, Y)$ is specified by

- The sequence of random proposals $Y$

$$Y = (y^{(0)}, y^{(1)}, \ldots, y^{(M-1)}), y^{(m)} \in \Omega, m = 0, \ldots, M-1$$

- The indices of the sites for the $M$ site visitations $j^{(0)}, j^{(1)}, \ldots, j^{(M-1)} \in \{1, \ldots, N_{\mathcal{D}}\}$

- The times of the $M$ site visitations $0 \leqslant t^{(0)} < t^{(1)} < \ldots < t^{(M-1)} \leqslant T$

---

**Algorithm 1** Core step of M-HMC

---

**Require:** $U$, potential for the target distribution $\pi$; $Q_i, i = 1, \ldots, N_{\mathcal{D}}$, single-site proposals; $k^{\mathcal{D}}$, kinetic energy for discrete component; $I(\cdot, \cdot, \cdot | x, U, K^{\mathcal{C}})$, reversible and volume-preserving integrator for continuous component; $\tau$, interval length in $\mathbb{T}^{N_{\mathcal{D}}}$

**input** $x^{(0)}$, discrete state; $q^{\mathcal{D}(0)}, p^{\mathcal{D}(0)}$, auxiliary location and momentum for discrete state; $q^{\mathcal{C}(0)}$, continuous location; $p^{\mathcal{C}(0)}$, auxiliary momentum for continuous state; $T$, travel time

**output** $x$, next discrete state; $q^{\mathcal{D}}, p^{\mathcal{D}}$, next auxiliary location and momentum for discrete state; $q^{\mathcal{C}}$, next continuous location; $p^{\mathcal{C}}$, next auxiliary momentum for continuous state

1: **function** M-HMC($x^{(0)}, q^{\mathcal{D}(0)}, p^{\mathcal{D}(0)}, q^{\mathcal{C}(0)}, p^{\mathcal{C}(0)}, T$)
2:    $x \leftarrow x^{(0)}, q^{\mathcal{D}} \leftarrow q^{\mathcal{D}(0)}, p^{\mathcal{D}} \leftarrow p^{\mathcal{D}(0)}$
3:    $q^{\mathcal{C}} \leftarrow q^{\mathcal{C}(0)}, p^{\mathcal{C}} \leftarrow p^{\mathcal{C}(0)}$
4:    $v_i \leftarrow (k^{\mathcal{D}})'(p_i^{\mathcal{D}}), i = 1, \ldots, N_{\mathcal{D}}$
5:    $t_i \leftarrow \frac{\tau(\text{sign}(v_i) + 1) - 2q_i^{\mathcal{D}}}{2v_i}, i = 1, \ldots, N_{\mathcal{D}}$
6:    **while** $T > 0$ **do**
7:       $j \leftarrow \text{argmin}_i \{t_i, i = 1, \ldots, N_{\mathcal{D}}\}$
8:       $\varepsilon = \min\{t_j, T\}$
9:       $q_i^{\mathcal{D}} \leftarrow q_i^{\mathcal{D}} + \varepsilon v_i, i = 1, \ldots, N_{\mathcal{D}}$
10:      $(q^{\mathcal{C}}, p^{\mathcal{C}}) \leftarrow I(q^{\mathcal{C}}, p^{\mathcal{C}}, \varepsilon | x, U, K^{\mathcal{C}})$
11:      $T \leftarrow T - \varepsilon$
12:      **if** $\varepsilon = t_j$ **then**
13:         $t_i \leftarrow t_i - t_j, i = 1, \ldots, N_{\mathcal{D}}$
14:         $\tilde{x} \sim Q_j(\cdot | x)$
15:         $\Delta E \leftarrow \log \frac{e^{-U(x, q^{\mathcal{C}})} Q_j(\tilde{x} | x)}{e^{-U(\tilde{x}, q^{\mathcal{C}})} Q_j(x | \tilde{x})}$
16:         **if** $k^{\mathcal{D}}(p_j^{\mathcal{D}}) > \Delta E$ **then**
17:            $x \leftarrow \tilde{x}, q_j^{\mathcal{D}} \leftarrow \tau - q_j^{\mathcal{D}}$
18:            $p_j^{\mathcal{D}} \leftarrow \text{sign}(p_j^{\mathcal{D}})(k^{\mathcal{D}})^{-1}(k^{\mathcal{D}}(p_j^{\mathcal{D}}) - \Delta E)$
19:            $v_j \leftarrow (k^{\mathcal{D}})'(p_j^{\mathcal{D}})$
20:         **else**
21:            $p_j^{\mathcal{D}} \leftarrow -p_j^{\mathcal{D}}, v_j \leftarrow -v_j$
22:         **end if**
23:         $t_j \leftarrow \frac{\tau(\text{sign}(v_j) + 1) - 2q_j^{\mathcal{D}}}{2v_j}$
24:      **end if**
25:    **end while**
26:    $E = U\left(x, q^{\mathcal{C}}\right) + K^{\mathcal{D}}(p^{\mathcal{D}}) + K^{\mathcal{C}}(p^{\mathcal{C}})$
27:    $E^{(0)} = U\left(x^{(0)}, q^{\mathcal{C}(0)}\right) + K^{\mathcal{D}}(p^{\mathcal{D}(0)}) + K^{\mathcal{C}}(p^{\mathcal{D}(0)})$
28:    **if** Uniform$([0, 1]) < e^{-(E - E^{(0)})}$ **then**
29:      $p^{\mathcal{D}} \leftarrow -p^{\mathcal{D}}, p^{\mathcal{C}} \leftarrow -p^{\mathcal{C}}$
30:    **else**
31:      $x \leftarrow x^{(0)}, q^{\mathcal{D}} \leftarrow q^{\mathcal{D}(0)}, p^{\mathcal{D}} \leftarrow p^{\mathcal{D}(0)}$
32:      $q^{\mathcal{C}} \leftarrow q^{\mathcal{C}(0)}, p^{\mathcal{C}} \leftarrow p^{\mathcal{C}(0)}$
33:    **end if**
34:    **return** $x, q^{\mathcal{D}}, p^{\mathcal{D}}, q^{\mathcal{C}}, p^{\mathcal{C}}$
35: **end function**

---

- The discrete states of the system at $M$ site visitations $x = x^{(0)}, x^{(1)}, \ldots, x^{(M-1)} \in \Omega$

- Accept/reject decisions for the $M$ site visitations $a^{(m)} = \mathbb{1}_{\{y^{(m)} = x^{(m+1)}\}}$, where $x^{(M)} = x'$

- The evolution of the location variables $q^{\mathcal{D}}(t), q^{\mathcal{C}}(t)$ and the momentum variables $p^{\mathcal{D}}(t), p^{\mathcal{C}}(t), 0 \leqslant t \leqslant T$. Note that we might have discontinuities in $p^{\mathcal{D}}(t)$. We use $p^{\mathcal{D}}(t^-)$ to denote the left limit and $p^{\mathcal{D}}(t^+)$ to denote the right limit.

**Countable number of probabilistic paths and decomposition of $R_T(s, B)$**

In order for a probabilistic path $\omega(s, T, Y)$ to be valid, the different components of $\omega(s, T, Y)$ have to interact with each other in a way as determined by the *M-HMC* function. For example, we should have $y_i^{(m)} = x_i^{(m)}, \forall i \neq j^{(m)}$ and

$$
x^{(m+1)} = \begin{cases} y^{(m)} & \text{if } k^{\mathcal{D}}(p^{\mathcal{D}}(t^{(m)-})) > \log \frac{\pi(x^{(m)}, q^{\mathcal{C}}(t^{(m)}))Q_{j^{(m)}}(y^{(m)}|x^{(m)})}{\pi(y^{(m)}, q^{\mathcal{C}}(t^{(m)}))Q_{j^{(m)}}(x^{(m)}|y^{(m)})} \\ x^{(m)} & \text{otherwise} \end{cases}
$$

For $s \in \Omega \times \Sigma$ and some given travel time $T$, we say a sequence of proposals $Y$ is compatible with $s, T$ and *M-HMC* if we can find a corresponding probabilistic path $\omega(s, T, Y)$ that's valid.

Not all sequences of proposals correspond to valid probabilistic paths. But even if we don't consider the compatibility of the sequence of proposals with $s, T$ and *M-HMC*, the set of all possible such sequences has only a countable number of elements. This is because we only need to look at sequences of finite length (because of the fixed travel time $T$), and all the individual proposals are on discrete state spaces with a finite number of states.

The above analysis indicates that for some starting point $s \in \Omega \times \Sigma$ and travel time $T$, running the *M-HMC* function would result in only a countable number of possible destinations $s'$. Furthermore, $\forall s, s' \in \Omega \times \Sigma$ for which $R_T(s, \{s'\}) > 0$, there are at most a countable number of probabilistic paths which bring $s$ to $s'$ in time $T$ through *M-HMC*.

Formally, given some travel time $T$ and a sequence of proposals $Y$, define

$$
\mathcal{D}(T, Y) = \{s \in \Omega \times \Sigma : Y \text{ is compatible with } s, T \text{ and } \textit{M-HMC}\}
$$

Use $\mathcal{T}_{T,Y} : \mathcal{D}(T, Y) \to \Omega \times \Sigma$ to denote the deterministic mapping defined by *M-HMC* (without the final accept/reject step) for the given $Y$ in time $T$ (so that $\mathcal{D}(T, Y)$ represents the domain of the mapping $\mathcal{T}_{T,Y}$), and use

$$
\mathcal{I}(T, Y) = \{s' \in \Omega \times \Sigma : \exists s \in \mathcal{D}(T, Y), s.t. \mathcal{T}_{T,Y}(s) = s'\}
$$

to denote the image of the mapping $\mathcal{T}_{T,Y}$. For a given $x \in \Omega$, use

$$
\mathcal{T}_{T,Y,x} : \{(q^{\mathcal{D}}, p^{\mathcal{D}}, q^{\mathcal{C}}, p^{\mathcal{C}}) \in \Sigma : s = (x, q^{\mathcal{D}}, p^{\mathcal{D}}, q^{\mathcal{C}}, p^{\mathcal{C}}) \in \mathcal{D}(T, Y)\} \to \Sigma
$$

to denote the deterministic mapping induced by $\mathcal{T}_{T,Y}$ on $\Sigma$. In other words,

$$
\forall s = (x, q^{\mathcal{D}}, p^{\mathcal{D}}, q^{\mathcal{C}}, p^{\mathcal{C}}) \in \mathcal{D}(T, Y), \mathcal{T}_{T,Y,x}((q^{\mathcal{D}}, p^{\mathcal{D}}, q^{\mathcal{C}}, p^{\mathcal{C}})) = (q^{\mathcal{D}\prime}, p^{\mathcal{D}\prime}, q^{\mathcal{C}\prime}, p^{\mathcal{C}\prime})
$$

where $s' = (x', q^{\mathcal{D}\prime}, p^{\mathcal{D}\prime}, q^{\mathcal{C}\prime}, p^{\mathcal{C}\prime}) = \mathcal{T}_{T,Y}(s)$. Define

$$
(\Omega \times \Sigma)(s, T) = \{s' = (x', q^{\mathcal{D}\prime}, p^{\mathcal{D}\prime}, q^{\mathcal{C}\prime}, p^{\mathcal{C}\prime}) \in \Omega \times \Sigma : R_T(s, \{s'\}) > 0\}
$$

$\forall s, s' \in \Omega \times \Sigma$ for which $R_T(s, \{s'\}) > 0$, further define

$$
\mathcal{P}(s, s', T) = \{Y \text{ a sequence of proposals}: s \in \mathcal{D}(T, Y) \text{ and } \mathcal{T}_{T,Y}(s) = s'\}
$$

Then both $(\Omega \times \Sigma)(s, T)$ and $\mathcal{P}(s, s', T)$ have at most a countable number of elements.

**Proof of detailed balance**

First, we note that it's trivially true that

$$
\varphi(s)R_T(s, \{s\}) = \varphi(s)R_T(s, \{s\}) \tag{1}
$$

Next, we consider $s' \neq s$. For a given travel time $T$ and a sequence of proposals $Y$, $\forall s \in \mathcal{D}(T, Y)$, we use $r_{T,Y}(s, s')$ to denote the probability of going from $s$ to $s'$ through the probabilistic path $\omega(s, T, Y)$. Since *M-HMC* (without the final accept/reject step) defines a deterministic mapping $\mathcal{T}_{T,Y}$ for given $T$ and $Y$, considering all $s' \neq s$, the only non-zero term is $r_{T,Y}(s, \mathcal{T}_{T,Y}(s))$. For all $s' \neq s, \mathcal{T}_{T,Y}(s)$, we have $r_{T,Y}(s, s') = 0$.

Using the above notation, $\forall s \in A$ and $B \subset \Omega \times \Sigma$ measurable for which $s \notin B$, we can write $R_T(s,B)$ as

$$R_T(s,B) = \sum_{s' \in B \cap (\Omega \times \Sigma)(s,T)} R_T(s,\{s'\})$$

$$= \sum_{s' \in B \cap (\Omega \times \Sigma)(s,T)} \sum_{Y \in \mathcal{P}(s,s',T)} r_{T,Y}(s,s')$$

$$= \sum_{s' \in B \cap (\Omega \times \Sigma)(s,T)} \sum_{Y \in \mathcal{P}(s,s',T)} r_{T,Y}(s,\mathcal{T}_{T,Y}(s))$$

For a given travel time $T$, $\forall s, s' \in \Omega \times \Sigma, s \neq s'$, if $R_T(s,\{s'\}) > 0$, then $\mathcal{P}(s,s',T) \neq \emptyset$. In Lemma 3, we prove that $\forall Y \in \mathcal{P}(s,s',T)$, the absolute value of the determinant of the Jacobian of $\mathcal{T}_{T,Y,x}$ is $|\det \mathcal{J} \mathcal{T}_{T,Y,x}| = 1$, for all $x \in \Omega$. Furthermore, the deterministic mapping $\mathcal{T}_{T,Y}$ is reversible, and there exists a sequence of proposals $\tilde{Y} \in \mathcal{P}(s',s,T)$, s.t. $s = \mathcal{T}_{T,Y}^{-1}(s') = \mathcal{T}_{T,\tilde{Y}}(s')$.

In Lemma 4, we prove that, $\forall s' = \mathcal{T}_{T,Y}(s) \neq s$,

$$\varphi(s)r_{T,Y}(s,s') = \varphi(s)r_{T,Y}(s,\mathcal{T}_{T,Y}(s)) = \varphi(s')r_{T,\tilde{Y}}(s',\mathcal{T}_{T,\tilde{Y}}(s')) = \varphi(s')r_{T,\tilde{Y}}(s',s)$$

Using the above results, it's not hard to see that, for the case where $A \cap B = \emptyset$,

$$\int_{\Sigma} \sum_{x \in A(\Theta)} R_T(s,B)\varphi(s)\mathrm{d}\Theta$$

$$= \int_{\Sigma} \sum_{x \in A(\Theta)} \sum_{s' \in B \cap (\Omega \times \Sigma)(s,T)} \sum_{Y \in \mathcal{P}(s,s',T)} r_{T,Y}(s,s')\varphi(s)\mathrm{d}\Theta$$

$$= \int_{\Sigma} \sum_{x \in A(\Theta)} \sum_{s' \in B \cap (\Omega \times \Sigma)(s,T)} \sum_{Y \in \mathcal{P}(s,s',T)} r_{T,\tilde{Y}}(s',s)\varphi(s')\mathrm{d}\Theta$$

$$\overset{\text{change of variables}}{=} \int_{\Sigma} \sum_{x' \in B(\Theta')} \sum_{s \in A \cap (\Omega \times \Sigma)(s',T)} \sum_{\tilde{Y} \in \mathcal{P}(s',s,T)} r_{T,\tilde{Y}}(s',s)\varphi(s')\frac{1}{|\det \mathcal{J} \mathcal{T}_{T,Y,x}|}\mathrm{d}\Theta'$$

$$= \int_{\Theta} \sum_{x' \in B(\Theta')} R_T(s',A)\varphi(s')\mathrm{d}\Theta'$$

Combining the above reasoning with Equation 1, the same result can be established for the case where $A \cap B \neq \emptyset$. This proves the desired detailed balance property of *M-HMC* w.r.t. $\varphi$

$$\int_{\Sigma} \sum_{x \in A(\Theta)} R_T((x,\Theta),B)\varphi((x,\Theta))\mathrm{d}\Theta = \int_{\Sigma} \sum_{x \in B(\Theta)} R_T((x,\Theta),A)\varphi((x,\Theta))\mathrm{d}\Theta$$

$\square$

### 1.2.2 Useful Lemmas

In this section, we prove a few useful lemmas to complete the proof of Theorem 1. W.l.o.g. we assume $\tau = 1$ in this section. The proof can be trivially modified to be applicable to arbitrary $\tau$.

First, we prove two lemmas, similar to Lemma 1 and Lemma 2 in Section 5.1 of [1].

**Lemma 1.** *(Refraction) Let* $\mathcal{T} : \mathbb{T}^{N_{\mathcal{D}}} \times \mathbb{R}^{N_{\mathcal{D}}} \to \mathbb{T}^{N_{\mathcal{D}}} \times \mathbb{R}^{N_{\mathcal{D}}}$ *be a transformation in* $\mathbb{T}^{N_{\mathcal{D}}}$ *that takes a unit mass located at* $q^{\mathcal{D}} = (q_1^{\mathcal{D}}, \ldots, q_{N_{\mathcal{D}}}^{\mathcal{D}})$ *and moves it with constant velocity* $v = ((k^{\mathcal{D}})'(p_1^{\mathcal{D}}), \ldots, (k^{\mathcal{D}})'(p_{N_{\mathcal{D}}}^{\mathcal{D}}))$. *Assume it reaches 0 or 1 at site j first. Subsequently* $q_j^{\mathcal{D}}$ *is changed to* $1 - q_j^{\mathcal{D}}$, *and* $p_j^{\mathcal{D}}$ *is changed to* $sign(p_j^{\mathcal{D}})(k^{\mathcal{D}})^{-1}(k^{\mathcal{D}}(p_j^{\mathcal{D}}) - \Delta E)$ *(where* $\Delta E$ *is a constant and satisfies* $\Delta E < k^{\mathcal{D}}(p_j^{\mathcal{D}})$). *The move is carried on, with the velocity* $v_j$ *changed to* $(k^{\mathcal{D}})'(sign(p_j^{\mathcal{D}})(k^{\mathcal{D}})^{-1}(k^{\mathcal{D}}(p_j^{\mathcal{D}}) - \Delta E))$, *for the total time period* $\mu$ *till it ends in location* $q^{\mathcal{D}'}$ *and momentum* $p^{\mathcal{D}'}$, *before it reaches 0 or 1 again at any sites. Then* $\mathcal{T}$ *is volume preserving, i.e. the absolute value of the determinant of its Jacobian* $|\det \mathcal{J} \mathcal{T}| = 1$.

*Proof.* Following the same argument as in the proof of Lemma 1 of [1], we have

$$|\det \mathcal{J}\mathcal{T}| = \left| \det \begin{pmatrix} \frac{\partial q^{\mathcal{D}_j\prime}}{\partial q_j^{\mathcal{D}}} & \frac{\partial q^{\mathcal{D}_j\prime}}{\partial p_j^{\mathcal{D}}} \\ \frac{\partial p^{\mathcal{D}_j\prime}}{\partial q_j^{\mathcal{D}}} & \frac{\partial p^{\mathcal{D}_j\prime}}{\partial p_j^{\mathcal{D}}} \end{pmatrix} \right|$$

If we define $t_j = \frac{\text{sign}(v_j) + 1 - 2q_j^{\mathcal{D}}}{2(k^{\mathcal{D}})\prime(p_j^{\mathcal{D}})} = \frac{\text{sign}(p_j^{\mathcal{D}}) + 1 - 2q_j^{\mathcal{D}}}{2(k^{\mathcal{D}})\prime(p_j^{\mathcal{D}})}$, then

$$
\begin{aligned}
p^{\mathcal{D}_j\prime} &= \text{sign}(p_j^{\mathcal{D}})(k^{\mathcal{D}})^{-1}(k^{\mathcal{D}}(p_j^{\mathcal{D}}) - \Delta E) \\
q^{\mathcal{D}_j\prime} &= \frac{1 - \text{sign}(p_j^{\mathcal{D}})}{2} + (k^{\mathcal{D}})\prime(p^{\mathcal{D}_j\prime})(\mu - t_j) \\
&= \frac{1 - \text{sign}(p_j^{\mathcal{D}})}{2} + (k^{\mathcal{D}})\prime(p^{\mathcal{D}_j\prime}) \left( \mu - \frac{\text{sign}(p_j^{\mathcal{D}}) + 1 - 2q_j^{\mathcal{D}}}{2(k^{\mathcal{D}})\prime(p_j^{\mathcal{D}})} \right)
\end{aligned}
$$

This implies

$$
\begin{aligned}
|\det \mathcal{J}\mathcal{T}| &= \left| \det \begin{pmatrix} \frac{\partial q^{\mathcal{D}_j\prime}}{\partial q_j^{\mathcal{D}}} & \frac{\partial q^{\mathcal{D}_j\prime}}{\partial p_j^{\mathcal{D}}} \\ \frac{\partial p^{\mathcal{D}_j\prime}}{\partial q_j^{\mathcal{D}}} & \frac{\partial p^{\mathcal{D}_j\prime}}{\partial p_j^{\mathcal{D}}} \end{pmatrix} \right| = \left| \det \begin{pmatrix} \frac{\partial q^{\mathcal{D}_j\prime}}{\partial q_j^{\mathcal{D}}} & \frac{\partial q^{\mathcal{D}_j\prime}}{\partial p_j^{\mathcal{D}}} \\ 0 & \frac{\partial p^{\mathcal{D}_j\prime}}{\partial p_j^{\mathcal{D}}} \end{pmatrix} \right| \\
&= \left| \frac{\partial q^{\mathcal{D}_j\prime}}{\partial q_j^{\mathcal{D}}} \frac{\partial p^{\mathcal{D}_j\prime}}{\partial p_j^{\mathcal{D}}} \right| = \left| \frac{(k^{\mathcal{D}})\prime(p^{\mathcal{D}_j\prime})}{(k^{\mathcal{D}})\prime(p_j^{\mathcal{D}})} \frac{(k^{\mathcal{D}})\prime(p_j^{\mathcal{D}})}{(k^{\mathcal{D}})\prime(p^{\mathcal{D}_j\prime})} \right| = 1
\end{aligned}
$$

$\square$

**Lemma 2.** *(Reflection) Let* $\mathcal{T} : \mathbb{T}^{N_{\mathcal{D}}} \times \mathbb{R}^{N_{\mathcal{D}}} \to \mathbb{T}^{N_{\mathcal{D}}} \times \mathbb{R}^{N_{\mathcal{D}}}$ *be a transformation in* $\mathbb{T}^{N_{\mathcal{D}}}$ *that takes a unit mass located at* $q^{\mathcal{D}} = (q_1^{\mathcal{D}}, \ldots, q_N^{\mathcal{D}})$ *and moves it with constant velocity* $v = ((k^{\mathcal{D}})\prime(p_1^{\mathcal{D}}), \ldots, (k^{\mathcal{D}})\prime(p_{N_{\mathcal{D}}}^{\mathcal{D}}))$. *Assume it reaches 0 or 1 at site j first. Subsequently* $p_j^{\mathcal{D}}$ *is changed to* $-p_j^{\mathcal{D}}$. *The move is carried on, with the velocity* $v_j$ *changed to* $-v_j$, *for the total time period* $\mu$ *till it ends in location* $q^{\mathcal{D}\prime}$ *and momentum* $p^{\mathcal{D}\prime}$, *before it reaches 0 or 1 at any sites again. Then* $\mathcal{T}$ *is volume preserving, i.e. the absolute value of the determinant of its Jacobian* $|\det \mathcal{J}\mathcal{T}| = 1$.

*Proof.* Following the same argument as in the proof of Lemma 2 of [1], we have

$$|\det \mathcal{J}\mathcal{T}| = \left| \det \begin{pmatrix} \frac{\partial q^{\mathcal{D}_j\prime}}{\partial q_j^{\mathcal{D}}} & \frac{\partial q^{\mathcal{D}_j\prime}}{\partial p_j^{\mathcal{D}}} \\ \frac{\partial p^{\mathcal{D}_j\prime}}{\partial q_j^{\mathcal{D}}} & \frac{\partial p^{\mathcal{D}_j\prime}}{\partial p_j^{\mathcal{D}}} \end{pmatrix} \right|$$

If we define $t_j = \frac{\text{sign}(v_j) + 1 - 2q_j^{\mathcal{D}}}{2(k^{\mathcal{D}})\prime(p_j^{\mathcal{D}})} = \frac{\text{sign}(p_j^{\mathcal{D}}) + 1 - 2q_j^{\mathcal{D}}}{2(k^{\mathcal{D}})\prime(p_j^{\mathcal{D}})}$, then

$$
\begin{aligned}
p^{\mathcal{D}_j\prime} &= -p_j^{\mathcal{D}} \\
q^{\mathcal{D}_j\prime} &= \frac{1 + \text{sign}(p_j^{\mathcal{D}})}{2} - (k^{\mathcal{D}})\prime(p_j^{\mathcal{D}})(\mu - t_j) \\
&= \frac{1 + \text{sign}(p_j^{\mathcal{D}})}{2} - (k^{\mathcal{D}})\prime(p_j^{\mathcal{D}}) \left( \mu - \frac{\text{sign}(p_j^{\mathcal{D}}) + 1 - 2q_j^{\mathcal{D}}}{2(k^{\mathcal{D}})\prime(p_j^{\mathcal{D}})} \right) \\
&= 1 + \text{sign}(p_j^{\mathcal{D}}) - (k^{\mathcal{D}})\prime(p_j^{\mathcal{D}})\mu - q_j^{\mathcal{D}}
\end{aligned}
$$

This implies

$$|\det \mathcal{J}\mathcal{T}| = \left| \det \begin{pmatrix} \frac{\partial q^{\mathcal{D}_j\prime}}{\partial q_j^{\mathcal{D}}} & \frac{\partial q^{\mathcal{D}_j\prime}}{\partial p_j^{\mathcal{D}}} \\ \frac{\partial p^{\mathcal{D}_j\prime}}{\partial q_j^{\mathcal{D}}} & \frac{\partial p^{\mathcal{D}_j\prime}}{\partial p_j^{\mathcal{D}}} \end{pmatrix} \right| = \left| \det \begin{pmatrix} -1 & \frac{\partial q^{\mathcal{D}_j\prime}}{\partial p_j^{\mathcal{D}}} \\ 0 & -1 \end{pmatrix} \right| = 1$$

$\square$

**Lemma 3.** *Given travel time* $T$, $\forall s, s' \in \Omega \times \Sigma, s \neq s'$ *for which* $R_T(s, \{s'\}) > 0$, $\mathcal{P}(s, s', T) \neq \emptyset$. $\forall Y \in \mathcal{P}(s, s', T)$, *the absolute value of the determinant of the Jacobian of* $\mathcal{T}_{T,Y,x}$ *is* $|\det \mathcal{J}\mathcal{T}_{T,Y,x}| = 1$, *for all* $x \in \Omega$ *where* $\mathcal{T}_{T,Y,x}$ *is well-defined. Furthermore, the deterministic mapping* $\mathcal{T}_{T,Y}$ *is reversible, and there exists a sequence of proposals* $\tilde{Y} \in \mathcal{P}(s', s, T)$, *s.t.* $s = \mathcal{T}_{T,Y}^{-1}(s') = \mathcal{T}_{T,\tilde{Y}}(s')$

*Proof.* Given travel time $T$, $\forall s, s' \in \Omega \times \Sigma$, if $R_T(s, \{s'\}) > 0$, then by definition $\mathcal{P}(s, s', T) \neq \emptyset$. $\forall Y \in \mathcal{P}(s, s', Y)$, for some $x \in \Omega$, if the deterministic mapping $\mathcal{T}_{T,Y,x}$ is well-defined, then $\mathcal{T}_{T,Y,x}$ can be be written as the composition of a sequence of deterministic mappings

$$\mathcal{T}_{T,Y,x} = \mathcal{T}_{T,Y,x}^{(0)} \circ \mathcal{T}_{T,Y,x}^{(1)} \circ \cdots \circ \mathcal{T}_{T,Y,x}^{(M-1)}$$

Each one of the mappings $\mathcal{T}_{T,Y,x}^{(m)}, m = 0, \ldots, M-1$ consists of two parts that don't interact: a discrete part that operates on $q^{\mathcal{D}}, p^{\mathcal{D}}$, and a continuous part that operates on $q^{\mathcal{C}}, p^{\mathcal{C}}$. The discrete part is either a refraction mapping as described in Lemma 1, or a reflection mapping as described in Lemma 2. The continuous part is given by the integrator $I$, which is reversible and volume-preserving. Using Lemma 1 and Lemma 2 and the properties of the integrator $I$, it's easy to see that the absolute value of the determinant of the Jacobian

$$|\det \mathcal{J} \mathcal{T}_{T,Y,x}| = \prod_{m=0}^{M-1} |\det \mathcal{J} \mathcal{T}_{T,Y,x}^{(m)}| = 1$$

$\forall Y \in \mathcal{P}(s, s', Y)$, define a new sequence of proposals $\tilde{Y} = (\tilde{y}^{(0)}, \tilde{y}^{(1)}, \ldots, \tilde{y}^{(M-1)})$ where

$$\tilde{y}^{(m)} = \begin{cases} x^{(M-m-1)} & \text{if } a^{(M-m-1)} = 1 \, (\text{i.e. } y^{(M-m-1)} = x^{(M-m)}) \\ y^{(M-m-1)} & \text{otherwise (i.e. } y^{(M-m-1)} \neq x^{(M-m)}, \text{which means } x^{(M-m-1)} = x^{(M-m)}) \end{cases}$$

We claim that $\tilde{Y} \in \mathcal{P}(s, s', T)$, and $\mathcal{T}_{T,\tilde{Y}}(s') = s$. To see $\tilde{Y}$ has these desired properties, we look at its corresponding probabilistic path $\omega(s', T, \tilde{Y})$. The corresponding discrete states of the system at $M$ site visitations $\tilde{x}^{(m)}, m = 0, \ldots, M$ and the indices of the sites for the $M$ site visitations $\tilde{j}^{(m)}, m = 0, \ldots, M-1$ are given by simple reversals of the original sequence of discrete states $x^{(m)}, m = 0, \ldots, M$ and the original sequence of indices for visited sites $j^{(m)}, m = 0, \ldots, M-1$:

$$\begin{aligned} \tilde{j}^{(m)} &= j^{(M-m-1)}, m = 0, \ldots, M-1 \\ \tilde{x}^{(m)} &= x^{(M-m)}, m = 0, \ldots, M \end{aligned}$$

The corresponding sequence of accept/reject decisions $\tilde{a}^{(m)}, m = 0, \ldots, M-1$ is also a simple reversal of the original sequence of accept/reject decisions $a^{(m)}, m = 0, \ldots, M-1$

$$\tilde{a}^{(m)} = \mathbb{1}_{\{\tilde{y}^{(m)} = \tilde{x}^{(m+1)}\}} = \begin{cases} \mathbb{1}_{\{x^{(M-m-1)} = x^{(M-m-1)}\}} = 1 & \text{if } a^{(M-m-1)} = 1 \\ \mathbb{1}_{\{y^{(M-m-1)} = x^{(M-m-1)}\}} = 0 & \text{if } a^{(M-m-1)} = 0 \end{cases} = a^{(M-m-1)}$$

It's straightforward to verify that $\omega(s', T, \tilde{Y})$ is a valid probabilistic path that brings $s'$ back to $s$ in time $T$ through *M-HMC*. In particular, note the importance of the momentum negating step in ensuring the existence of such a probabilistic path. This proves our claim.

$\square$

**Lemma 4.** $\forall s, s' \in \Omega \times \Sigma, s \neq s'$ *for which* $R_T(s, \{s'\}) > 0$, *for* $Y \in \mathcal{P}(s, s', T)$, *we have*

$$\varphi(s) r_{T,Y}(s, s') = \varphi(s) r_{T,Y}(s, \mathcal{T}_{T,Y}(s)) = \varphi(s') r_{T,\tilde{Y}}(s', \mathcal{T}_{T,\tilde{Y}}(s')) = \varphi(s') r_{T,\tilde{Y}}(s', s)$$

*where* $\tilde{Y}$ *is defined as in Lemma 3.*

*Proof.* We can directly calculate the transition probability $r_{T,Y}(s, s')$. Define

$$E = U(x, q^{\mathcal{C}}) + K^{\mathcal{D}}(p^{\mathcal{D}}) + K^{\mathcal{C}}(p^{\mathcal{C}}), E' = U(x', q^{\mathcal{C}'}) + K^{\mathcal{D}}(p^{\mathcal{D}'}) + K^{\mathcal{C}}(p^{\mathcal{C}'})$$

Then

$$r_{T,Y}(s, s') = \prod_{m=0}^{M-1} Q_{j^{(m)}}(y^{(m)} | x^{(m)}) \min\{1, e^{-(E'-E)}\}$$

Correspondingly, we can also calculate the transition probability $r_{T,\tilde{Y}}(s', s)$.

$$r_{T,\tilde{Y}}(s', s) = \prod_{m=0}^{M-1} Q_{\tilde{j}^{(m)}}(\tilde{y}^{(m)} | \tilde{x}^{(m)}) \min\{1, e^{-(E-E')}\}$$

Note that

$$
\begin{aligned}
\frac{r_{T,Y}(s,s')}{\min\{1, e^{-(E'-E)}\}} &= \prod_{m=0}^{M-1} Q_{j^{(m)}}^{a^{(m)}}(y^{(m)}|x^{(m)}) \prod_{m=0}^{M-1} Q_{j^{(m)}}^{1-a^{(m)}}(y^{(m)}|x^{(m)}) \\
&= \prod_{m:a^{(m)}=1} Q_{j^{(m)}}(y^{(m)}|x^{(m)}) \prod_{m:a^{(m)}=0} Q_{j^{(m)}}(y^{(m)}|x^{(m)}) \\
\frac{r_{T,\tilde{Y}}(s',s)}{\min\{1, e^{-(E-E')}\}} &= \prod_{m=0}^{M-1} Q_{\tilde{j}^{(m)}}^{\tilde{a}^{(m)}}(\tilde{y}^{(m)}|\tilde{x}^{(m)}) \prod_{m=0}^{M-1} Q_{\tilde{j}^{(m)}}^{1-\tilde{a}^{(m)}}(\tilde{y}^{(m)}|\tilde{x}^{(m)}) \\
&= \prod_{m:\tilde{a}^{(m)}=1} Q_{\tilde{j}^{(m)}}(\tilde{y}^{(m)}|\tilde{x}^{(m)}) \prod_{m:\tilde{a}^{(m)}=0} Q_{\tilde{j}^{(m)}}(\tilde{y}^{(m)}|\tilde{x}^{(m)}) \\
&= \prod_{m:a^{(M-m-1)}=1} Q_{j^{(M-m-1)}}(x^{(M-m-1)}|y^{(M-m-1)}) \\
&\quad\times \prod_{m:a^{(M-m-1)}=0} Q_{j^{(M-m-1)}}(y^{(M-m-1)}|x^{(M-m)}) \\
&= \prod_{m:a^{(M-m-1)}=1} Q_{j^{(M-m-1)}}(x^{(M-m-1)}|y^{(M-m-1)}) \\
&\quad\times \prod_{m:a^{(M-m-1)}=0} Q_{j^{(M-m-1)}}(y^{(M-m-1)}|x^{(M-m-1)}) \\
&= \prod_{m:a^{(m)}=1} Q_{j^{(m)}}(x^{(m)}|y^{(m)}) \prod_{m:a^{(m)}=0} Q_{j^{(m)}}(y^{(m)}|x^{(m)})
\end{aligned}
$$

By following the probabilistic path $\omega(s, T, Y)$ and doing explicit calculations, we can show that

$$
\varphi(s) r_{T,Y}(s,s') = \varphi(s') r_{T,\tilde{Y}}(s',s)
$$

□

## 2 Details on implementation with Laplace momentum

---
**Algorithm 2** Definition of *GetStepSizesNSteps*

---
1: **function** GetStepSizesNSteps($\varepsilon, T, L, N_{\mathcal{D}}, n_{\mathcal{D}}$)
2: $\quad \Phi \sim \text{Dirichlet}_{N_{\mathcal{D}}+1}(1)$; $\Phi_1 \leftarrow \Phi_1 + \Phi_{N_{\mathcal{D}}+1}$
3: $\quad \eta_t \leftarrow \sum_{s=1}^{n_{\mathcal{D}}} \Phi_{[(t-1)n_{\mathcal{D}}+s] \bmod N_{\mathcal{D}}}, t = 1, \ldots, L$; $\eta_1 \leftarrow \eta_1 - \Phi_{N_{\mathcal{D}}+1}$
4: $\quad \eta_t \leftarrow T\eta_t / \sum_{s=1}^{L} \eta_s, t = 1, \ldots, L$; $M_t \leftarrow \lceil \eta_t/\varepsilon \rceil, t = 1, \ldots, L$; $\eta_t \leftarrow \eta_t/M_t, t = 1, \ldots, L$
5: $\quad$ **return** $\eta, M$
6: **end function**

---

In what follows, line numbers refer to lines in Algorithm 1. Under Laplace momentum, $v_i = \text{sign}(p_i^{\mathcal{D}}) \in \{1, -1\}$. As a result, different $q_i^{\mathcal{D}}$ always evolve with a constant speed 1, and we no longer need the argmin in Line 7. Site visitation order is completely determined by the initial sampling of $q^{\mathcal{D}}, p^{\mathcal{D}}$. Furthermore, we can precompute all the involved step sizes (in Line 8). These step sizes are in fact differences of neighboring order statistics of $N^{\mathcal{D}}$ uniform samples on $[0, \tau]$, and as a result have the Dirichlet distribution as the joint distribution. The initial momentum is given by $p_i^{\mathcal{D}(0)} \sim \nu(p) \propto e^{-|p|}$, which corresponds to the initial kinetic energy $k^{\mathcal{D}}(p_i^{\mathcal{D}(0)}) \sim \text{Exponential}(1)$.

The above observations indicate that, using Laplace momentum, we no longer need to keep track of $q^{\mathcal{D}}, p^{\mathcal{D}}$. Instead, at the beginning of each iteration, we can sample the site visitation order as a random permutation, the step sizes from a Dirichlet distribution, and the kinetic energies from independent exponential distributions. In each iteration, we simply evolve the system according to the step sizes, visit each site in order, and keep track of changes in kinetic energies. These simplications results in the efficient implementation described in Algorithm 1 in the main text. See also Algorithm 2 for the definition of the function *GetStepSizesNSteps* in Algorithm 1 in the main text.

# 3 Python function for comparing M-HMC with naive MH within HMC

Code for reproducing the results in the paper is available at `https://github.com/StannisZhou/mixed_hmc`. In particular, we include below a illustrative python function for comparing M-HMC with naive Metropolis updates within HMC. Experimental results using this function can be reproduced using the script *test_naive_mixed_hmc.py* under *scripts/simple_gmm*.

```python
import numpy as np

import numba
from tqdm import tqdm

def naive_mixed_hmc(x0, q0, n_samples, epsilon, L, pi, mu_list, sigma_list, use_k=True):
    """Function for comparing mixed HMC and naive Metropolis updates within HMC

    Parameters
    ----------
    x0 : int
        Discrete variable for the mixture component
    q0 : float
        Continuous variable for the state of GMM
    n_samples : int
        Number of samples to draw
    epsilon : float
        Step size
    L : int
        Number of steps
    pi : np.array
        Array of shape (n_components,). The probabilities for different components
    mu_list : np.array
        Array of shape (n_components,). Means of different components
    sigma_list : np.array
        Array of shape (n_components,). Standard deviations of different components
    use_k : bool
        True if we use mixed HMC. False if we make naive Metropolis updates within HMC

    Returns
    -------
    x_samples : np.array
        Array of shape (n_samples,). Samples for x
    q_samples : np.array
        Array of shape (n_samples,). Samples for x
    accept_list : np.array
        Array of shape (n_samples,). Records whether we accept or reject at each step
    """

    @numba.jit(nopython=True)
    def potential(x, q):
        potential = (
            -np.log(pi[x])
            + 0.5 * np.log(2 * np.pi * sigma_list[x] ** 2)
            + 0.5 * (q - mu_list[x]) ** 2 / sigma_list[x] ** 2
        )
        return potential

    @numba.jit(nopython=True)
    def grad_potential(x, q):
        grad_potential = (q - mu_list[x]) / sigma_list[x] ** 2
```

```python
        return grad_potential

@numba.jit(nopython=True)
def take_naive_mixed_hmc_step(x0, q0, epsilon, L, n_components):
    # Resample momentum
    p0 = np.random.randn()
    k0 = np.random.exponential()
    # Initialize q, k
    x = x0
    q = q0
    p = p0
    k = k0
    # Take L steps
    for ii in range(L):
        q, p = leapfrog_step(x=x, q=q, p=p, epsilon=epsilon)
        x, k = update_discrete(x0=x, k0=k, q=q, n_components=n_components)

    # Accept or reject
    current_U = potential(x0, q0)
    current_K = k0 + 0.5 * p0 ** 2
    proposed_U = potential(x, q)
    proposed_K = k + 0.5 * p ** 2
    accept = np.random.rand() < np.exp(
        current_U - proposed_U + current_K - proposed_K
    )
    if not accept:
        x, q = x0, q0

    return x, q, accept

@numba.jit(nopython=True)
def leapfrog_step(x, q, p, epsilon):
    p -= 0.5 * epsilon * grad_potential(x, q)
    q += epsilon * p
    p -= 0.5 * epsilon * grad_potential(x, q)
    return q, p

@numba.jit(nopython=True)
def update_discrete(x0, k0, q, n_components):
    x = x0
    k = k0
    distribution = np.ones(n_components)
    distribution[x] = 0
    distribution /= np.sum(distribution)
    proposal_for_ind = np.argmax(np.random.multinomial(1, distribution))
    x = proposal_for_ind
    delta_E = potential(x, q) - potential(x0, q)
    # Decide whether to accept or reject
    if use_k:
        accept = k > delta_E
        if accept:
            k -= delta_E
        else:
            x = x0
    else:
        accept = np.random.exponential() > delta_E
        assert k == k0
        if not accept:
            x = x0
```

```
        return x, k

    x, q = x0, q0
    x_samples, q_samples, accept_list = [], [], []
    for _ in tqdm(range(n_samples)):
        x, q, accept = take_naive_mixed_hmc_step(
            x0=x, q0=q, epsilon=epsilon, L=L, n_components=pi.shape[0]
        )
        x_samples.append(x)
        q_samples.append(q)
        accept_list.append(accept)

    x_samples = np.array(x_samples)
    q_samples = np.array(q_samples)
    accept_list = np.array(accept_list)
    return x_samples, q_samples, accept_list
```

## 4  Binary HMC Samplers are special cases of M-HMC

Formally, we have the following equivalence between binary HMC and M-HMC:

**Proposition 1.** *Binary HMC is equivalent to a variant of M-HMC (where $q^{\mathcal{D}}$ is initialized at the start and not resampled at each iteration) with $\tau = 1$ and deterministic proposals $Q_i, i = 1, \ldots, N_{\mathcal{D}}$*

$$Q_i(\tilde{x}|x) = \begin{cases} 1, \textit{ if } \tilde{x}_i = -x_i, \tilde{x}_j = x_j, \forall j \neq i \\ 0, \textit{ otherwise} \end{cases}$$

*Gaussian and exponential binary HMC correspond to $k^{\mathcal{D}}(p) = |p|$ and $k^{\mathcal{D}}(p) = |p|^{\frac{2}{3}}$ respectively.*

Since no continuous component is involved in a binary distribution, for notational simplicity, we drop all the superscript $\mathcal{D}$ in the following discussions. We consider the family of kinetic energies $K_\beta(p) = |p|^\beta$, and define the corresponding distribution to be $\nu_\beta(p) \propto e^{-K_\beta(p)}$. We want to show that the binary HMC samplers are special cases of a variant of M-HMC. In what follows, we use M-HMC to refer to the variant of M-HMC where $q$ is initialized at the start and not resampled at each iteration.

In order to establish the equivalence between binary HMC and M-HMC, we need to study:

1. For site $j$, the distribution on the initial time it takes to visit site $j$, which we denote by $t_j^{(0)}$.
   - As shown in Algorithm 1, in M-HMC
     $$t_j^{(0)} = \frac{\operatorname{sign}(v_j^{(0)}) + 1 - 2q_j^{(0)}}{2v_j^{(0)}}$$
     where $v_j^{(0)} = K'_\beta(p_j^{(0)}) = \operatorname{sign}(p_j^{(0)})\beta|p_j^{(0)}|^{\beta-1}$ is the velocity at site $j$, and
     $$q_j^{(0)} \sim U([0,1]), p_j^{(0)} \sim \nu_\beta(p_j^{(0)})$$
   - For the Gaussian binary HMC sampler,
     $$t_j^{(0)} = \begin{cases} -\arctan\left(\dfrac{q_j^{(0)}}{p_j^{(0)}}\right) & \text{if } \dfrac{q_j^{(0)}}{p_j^{(0)}} \leqslant 0 \\ \pi - \arctan\left(\dfrac{q_j^{(0)}}{p_j^{(0)}}\right) & \text{if } \dfrac{q_j^{(0)}}{p_j^{(0)}} > 0 \end{cases}$$
     where $q_j^{(0)}, p_j^{(0)} \sim N(0,1)$.
   - For the exponential binary HMC sampler,
     $$t_j^{(0)} = p_j^{(0)} + \sqrt{(p_j^{(0)})^2 + 2q_j^{(0)}}$$
     where $q_j^{(0)} \sim \exp(1), p_j^{(0)} \sim N(0,1)$.

2. For site $j$, the distribution on the initial total energy, which we denote by $k_j^{(0)}$.

- For M-HMC, $k_j^{(0)} = K_\beta(p_j^{(0)})$, where $p_j^{(0)} \sim \nu_\beta(p_j^{(0)})$.
- For the Gaussian binary HMC sampler,

$$k_j^{(0)} = \frac{1}{2}(q_j^{(0)})^2 + \frac{1}{2}(p_j^{(0)})^2$$

where $q_j^{(0)}, p_j^{(0)} \sim N(0,1)$.
- For the exponential binary HMC sampler,

$$k_j^{(0)} = q_j^{(0)} + \frac{1}{2}(p_j^{(0)})^2$$

where $q_j^{(0)} \sim \exp(1), p_j^{(0)} \sim N(0,1)$.

3. For site $j$, after we reach 0 or 1, if we have total energy $k$, the time it takes to hit a boundary again at this site. We denote this time by $t_j(k)$.

- For M-HMC, $t_j(k) = \frac{1}{\beta k^{1-\frac{1}{\beta}}}$
- For the Gaussian binary HMC, $t_j(k) = \pi$
- For the exponential binary HMC, $t_j(k) = 2\sqrt{2k}$

Since different dimensions are independent of each other, we only need to look at one particular dimension $j$. We can prove the corresponding propositions if we can establish suitable equivalence concerning the joint distribution on $(t_j^{(0)}, k_j^{(0)})$, and the function $t_j(k)$.

## 4.1 Proof of Proposition 1 for Gaussian binary HMC

In order to prove Proposition 1 for Gaussian binary HMC, we first prove a lemma

**Lemma 5.** *Assume $q, p \sim N(0,1)$ are two independent standard normal random variables. Then $\frac{q}{p}$ and $q^2 + p^2$ are independent. Furthermore,* $\arctan\left(\frac{q}{p}\right)$ *follows the uniform distribution* $U\left(\left[-\frac{\pi}{2}, \frac{\pi}{2}\right]\right)$, *and* $\frac{q^2+p^2}{2}$ *follows the exponential distribution* $\exp(1)$.

*Proof.* We calculate the characteristic function of the random vector $\left(\frac{q}{p}, q^2 + p^2\right)$:

$$
\begin{aligned}
& \mathbb{E}_{q,p \sim N(0,1)}\left[e^{i\left[t_1\frac{q}{p}+t_2(q^2+p^2)\right]}\right] \\
= & \frac{1}{2\pi}\int_{\mathbb{R}^2} e^{it_1\frac{q}{p}+it_2(q^2+p^2)}e^{-\frac{q^2+p^2}{2}}\,\mathrm{d}q\mathrm{d}p \\
= & \frac{1}{2\pi}\int_0^{+\infty}\int_0^{2\pi} e^{it_1\tan\theta}e^{it_2r^2}e^{-\frac{r^2}{2}}r\,\mathrm{d}r\mathrm{d}\theta \\
= & \left[\int_0^{2\pi} e^{it_1\tan\theta}\frac{1}{2\pi}\mathrm{d}\theta\right]\left[\int_0^{+\infty} e^{it_2r^2-\frac{r^2}{2}}r\,\mathrm{d}r\right] \\
= & \left[\int_{-\frac{\pi}{2}}^{\frac{\pi}{2}} e^{it_1\tan\theta}\frac{1}{\pi}\mathrm{d}\theta\right]\left[\int_0^{+\infty} e^{it_2x}\frac{1}{2}e^{-2x}\mathrm{d}x\right] \\
= & \left[\int_{-\infty}^{+\infty} e^{it_1x}\frac{1}{\pi(1+x^2)}\mathrm{d}x\right]\left[\int_0^{+\infty} e^{it_2x}\frac{1}{2}e^{-2x}\mathrm{d}x\right] \\
= & \mathbb{E}_{x\sim\text{Cauchy}(0,1)}[e^{it_1x}]\mathbb{E}_{x\sim\exp(2)}[e^{it_2x}]
\end{aligned}
$$

This calculation implies that $\frac{q}{p}$ and $q^2 + p^2$ are independent, and that $\frac{q}{p} \sim \text{Cauchy}(0,1)$, $q^2 + p^2 \sim \exp(2)$. Since the cumulative distribution function (CDF) of $\text{Cauchy}(0,1)$ is given by

$$\frac{1}{\pi}\arctan(x) + \frac{1}{2}$$

we have $\frac{1}{\pi}\arctan\left(\frac{q}{p}\right)+\frac{1}{2} \sim U([0,1])$, which implies that $\arctan\left(\frac{q}{p}\right) \sim U\left(\left[-\frac{\pi}{2},\frac{\pi}{2}\right]\right)$. From $q^2+p^2 \sim \exp(2)$, it's easy to deduce that $\frac{q^2+p^2}{2} \sim \exp(1)$. $\qquad\square$

*Proof.* **(Proposition 1 for Gaussian binary HMC)** For the Gaussian binary HMC sampler, using Lemma 5 and the expressions we derived in Section 4, given a dimension $j$, it's easy to see that $t_j^{(0)}$ and $k_j^{(0)}$ are independent, and that $t_j^{(0)} \sim U([0,\pi])$, $k_j^{(0)} \sim \exp(1)$. For M-HMC with $\beta = 1$, it's easy to see that we also have $t_j^{(0)}$ and $k_j^{(0)}$ are independent, and that $t_j^{(0)} \sim U([0,1])$, $k_j^{(0)} \sim \exp(1)$. This implies that the random vector $\left(\frac{t_j^{(0)}}{\pi}, k_j^{(0)}\right)$ from the Gaussian binary HMC sampler has the same joint distribution as the random vector $(t_j^{(0)}, k_j^{(0)})$ from M-HMC with $\beta = 1$.

For the Gaussian binary HMC sampler, $t_j(k) = \pi$, which is a constant function and is independent of the value of $k$. For M-HMC with $\beta = 1$, it's easy to see that $t_j(k) = 1$, which is also a constant function. This implies that $\forall k$, $\frac{t_j(k)}{\pi}$ for the Gaussian binary HMC sampler is equivalent to $t_j(k)$ for M-HMC with $\beta = 1$.

The above equivalences imply that the Gaussian binary HMC has exactly the same behavior as M-HMC with $\beta = 1$. In fact, the Gaussian binary HMC sampler behaves like scaling the time of M-HMC with $\beta = 1$ by $\pi$. $\qquad\square$

## 4.2 Proof of Proposition 1 for exponential binary HMC

*Proof.* **(Proposition 1 for exponential binary HMC)** Using the expressions we derived in Section 4, we can see that, at a given site $j$,

- For the exponential binary HMC sampler, the joint distribution of the random vector $(t_j^{(0)}, k_j^{(0)})$ is the same as the random vector $\left(p + \sqrt{p^2+2q}, q + \frac{1}{2}p^2\right)$, where $q \sim \exp(1), p \sim N(0,1)$ are independent. For a given total energy level $k$, $t_j(k) = 2\sqrt{2k}$.

- For M-HMC with $\beta = \frac{2}{3}$, the joint distribution of the random vector $(t_j^{(0)}, k_j^{(0)})$ is the same as the random vector $\left(\frac{3}{2}q|p|^{\frac{1}{3}}, |p|^{\frac{2}{3}}\right)$, where $q \sim U([0,1]), p \sim G\left(0,1,\frac{2}{3}\right)$ are independent. For a given total energy level $k$, $t_j(k) = \frac{3}{2}\sqrt{k}$.

In order to establish the equivalence between these two samplers, we calculate the characteristic functions of two random vectors. We first calculate the characteristic function of the random vector $\left(p + \sqrt{p^2+2q}, q + \frac{1}{2}p^2\right)$, where $q \sim \exp(1), p \sim N(0,1)$ are independent:

$$\mathbb{E}_{q\sim\exp(1),p\sim N(0,1)}\left[e^{i\left[t_1\left(p+\sqrt{p^2+2q}\right)+t_2\left(q+\frac{1}{2}p^2\right)\right]}\right]$$

$$= \frac{1}{\sqrt{2\pi}}\int_0^{+\infty}\int_{\mathbb{R}} e^{it_1\left(p+\sqrt{p^2+2q}\right)+it_2\left(q+\frac{p^2}{2}\right)}e^{-q}e^{-\frac{p^2}{2}}\,\mathrm{d}p\mathrm{d}q$$

$$= \frac{1}{2\sqrt{2\pi}}\int_{\mathbb{R}^2} e^{it_1\left(p+\sqrt{p^2+2|q|}\right)+it_2\left(|q|+\frac{p^2}{2}\right)}e^{-|q|}e^{-\frac{p^2}{2}}\,\mathrm{d}p\mathrm{d}q$$

$$\overset{p=r\cos\theta,q=\text{sign}(\sin\theta)\frac{r^2\sin^2\theta}{2}}{=} \frac{1}{2\sqrt{2\pi}}\int_0^{+\infty}\int_0^{2\pi} e^{it_1 r(1+\cos\theta)+it_2\frac{r^2}{2}}e^{-\frac{r^2}{2}}r^2\sin\theta\mathrm{d}\theta\mathrm{d}r$$

Next we calculate the characteristic function of the random vector $\left(2\sqrt{2}q|p|^{\frac{1}{3}}, |p|^{\frac{2}{3}}\right)$, where $q \sim U([0,1]), p \sim G\left(0,1,\frac{2}{3}\right)$ are independent:

$$\mathbb{E}_{q\sim U([0,1]),p\sim G\left(0,1,\frac{2}{3}\right)}\left[e^{i\left(t_1 2\sqrt{2}q|p|^{\frac{1}{3}}+t_2|p|^{\frac{2}{3}}\right)}\right]$$

$$= \frac{\frac{2}{3}}{2\Gamma\left(\frac{3}{2}\right)}\int_0^1\int_{\mathbb{R}} e^{it_1 2\sqrt{2}q|p|^{\frac{1}{3}}+it_2|p|^{\frac{2}{3}}}e^{-|p|^{\frac{2}{3}}}\,\mathrm{d}p\mathrm{d}q$$

$$= \frac{2}{3\sqrt{\pi}}\int_0^1\int_{\mathbb{R}} e^{it_1 2\sqrt{2}q|p|^{\frac{1}{3}}+it_2|p|^{\frac{2}{3}}}e^{-|p|^{\frac{2}{3}}}\,\mathrm{d}p\mathrm{d}q$$

$$= \frac{4}{3\sqrt{\pi}}\int_0^1\int_0^{+\infty} e^{it_1 2\sqrt{2}qp^{\frac{1}{3}}+it_2 p^{\frac{2}{3}}}e^{-p^{\frac{2}{3}}}\,\mathrm{d}p\mathrm{d}q$$

$$\overset{q=\frac{1+\cos\theta}{2},p=\frac{r^3}{2^{\frac{3}{2}}}}{=} \frac{4}{3\sqrt{\pi}}\int_0^{\pi}\int_0^{+\infty} e^{it_1 r(1+\cos\theta)+it_2\frac{r^2}{2}}e^{-\frac{r^2}{2}}\frac{3}{2^{\frac{5}{2}}}r^2\sin\theta\mathrm{d}r\mathrm{d}\theta$$

$$= \frac{1}{\sqrt{2\pi}}\int_0^{+\infty}\left[\int_0^{\pi} e^{it_1 r(1+\cos\theta)}\sin\theta\mathrm{d}\theta\right]e^{it_2\frac{r^2}{2}-\frac{r^2}{2}}r^2\mathrm{d}r$$

$$= \frac{1}{2\sqrt{2\pi}}\int_0^{+\infty}\left[\int_0^{2\pi} e^{it_1 r(1+\cos\theta)}\sin\theta\mathrm{d}\theta\right]e^{it_2\frac{r^2}{2}-\frac{r^2}{2}}r^2\mathrm{d}r$$

$$= \frac{1}{2\sqrt{2\pi}}\int_0^{+\infty}\int_0^{2\pi} e^{it_1 r(1+\cos\theta)+it_2\frac{r^2}{2}}e^{-\frac{r^2}{2}}r^2\sin\theta\mathrm{d}\theta\mathrm{d}r$$

The above calculations indicate that the joint distribution of $(t_j^{(0)}, k_j^{(0)})$ for the exponential binary HMC sampler is equivalent to the joint distribution of $\left(\frac{4\sqrt{2}}{3}t_j^{(0)}, k_j^{(0)}\right)$ for M-HMC with $\beta = \frac{2}{3}$. Furthermore, if we multiply the $t_j(k)$ function of M-HMC with $\beta = \frac{2}{3}$ by $\frac{4\sqrt{2}}{3}$, we get the function $2\sqrt{2k}$, which is exactly the $t_j(k)$ function for the exponential binary HMC sampler.

The above equivalences imply that the exponential binary HMC has exactly the same behavior as M-HMC with $\beta = \frac{2}{3}$. In fact, the exponential binary HMC sampler behaves like scaling the time of M-HMC with $\beta = \frac{2}{3}$ by $\frac{3}{4\sqrt{2}}$. $\qquad\square$

# 5 Some more details on numerical experiments

## 5.1 Exact parameter values for different samplers for 24D GMM

NUTS and NwG require no manual tuning. We favor HwG and DHMC by doing a parameter grid search and pick the setting with best MRESS for $x$, resulting in step size 1.1 and number of steps 80 for HwG, and a step-size range $(0.8, 1.0)$ and a number-of-steps range $(30, 40)$ for DHMC. We tune M-HMC by conducting short trial runs and inspecting the acceptance probabilities and traceplots, resulting in $\varepsilon = 1.7, L = 80, T = 136, n_{\mathcal{D}} = 1$.

## 5.2 Some additional CTM results

We also inspect traceplots and samples histograms of posterior samples for $\eta_{d1}$ on a document where Gibbs agrees with the other 3 samplers (Figure 1. NwG is excluded since it behaves similarly to HwG but is less efficient). The conclusions are similar to those in Section 3.3 of the main text: M-HMC clearly mixes the fastest, with HwG also outperforming Gibbs. Moreover, HwG and M-HMC explore the state space much more thoroughly.

# References

[1] Hadi Mohasel Afshar and Justin Domke. Reflection, refraction, and hamiltonian monte carlo. In C Cortes, N D Lawrence, D D Lee, M Sugiyama, and R Garnett, editors, *Advances in Neural Information Processing Systems 28*, pages 3007–3015. Curran Associates, Inc., 2015.

Figure 1: Traceplots and samples histograms of posterior samples of $\eta_{d1}$ on a document where Gibbs agrees with HwG, NwG&M-HMC in posterior means for $\eta_{d1}$