[Reviews · NeurIPS 2020]

Review 1

Summary and Contributions: The paper proposes an extension of Hamiltonian Monte Carlo that can handle a mixture of continuous and discrete variables. The main idea is to introduce auxiliary continuous variables, one per discrete variable, each of them taking values from a fixed interval, which is wrapped circularly and crossing the boundary on its ends triggers resampling of the corresponding discrete variable, subject to a reflect/refract step. The method, called M-HMC, can be used with any kinetic energy functions, although the authors in particular recommend the Laplace momentum. The paper contains a comprehensive comparison with alternative variants of HMC. Automatic tuning of step sizes, like in the no-U-turn sampler, is left for future work. EDIT: Not much to add after rebuttal. I still think it's a good paper and I found the authors' answers to the concerns raised by other reviewers satisfactory.

Strengths: HMC has seen a lot of success in various probabilistic programming systems, most prominently Stan, in many cases enabling essentially black-box inference. Unfortunately, its inability to handle discrete variables precludes the use of many popular models in a similar fashion. The problem being addressed is therefore an important one, and the work is potentially going to be very influential. The derivations are presented carefully and the algorithm appears correct, although I have not checked it in detail. The rationale for the design of the algorithm makes sense and the empirical evaluation is very extensive.

Weaknesses: The numerical experiments section is rather dense and it's not very clear how much of a benefit the proposed method ultimately offers over the baselines. A concise summary of what the results show would be very useful.

Correctness: The method appears correct and the experiments are sound.

Clarity: The paper is clear but rather dry. It took some willpower to go through it.

Relation to Prior Work: The work is clearly placed in the context of existing approaches to the same problem, explaining their shortcomings and highlighting its novelty.

Reproducibility: Yes

Additional Feedback:


Review 2

Summary and Contributions: The paper proposed a new variant of HMC called M-HMC that can work on space with mixed discrete and continuous support. The main motivation that makes it better is to update discrete and continuous variables more frequently, compared to previous HMC variants like PPHMC and DHMC. This is achieved by replacing the discrete variables by auxiliary position variables on a flat torus with the same dimension as discrete variables, and associating an "always-moving-away" MH step for the discrete variables. This position variable and the original continuous variable together with corresponding momentum variables forms a Hamiltonian system which can be integrated to propose the next state. During the integration, the values of the original discrete variables are explicitly recorded, as they are not coupled with the auxiliary position variables. The paper also relates this variant to DHMC and Gibbs-in-HMC and showing them are special case of M-HMC. With three experiments, the paper demonstrated a much better performance to baselines methods, including DHMC, NUTS-within-Gibbs and pure NUTS via integrating out the discrete variables.

Strengths: The paper addresses an important problem for HMC and leads to a method with potential wide usage. The relation of the proposed methods and prior works has been discussed as well to explain why M-HMC is better. Well-designed experiments have been done to show empirical performance gain of the proposed method, and how the method performs with different hyper-parameters. Necessary theoretical work on the validity of M-HMC is done as well, which is great.

Weaknesses: I have a few concerns for the experiments ## How HMC-within-Gibbs is used as a baseline 1. The concrete variant used is NUTS-within-Gibbs. In fact, it might be worth checking just HMC-within-Gibbs because the no-U-turn criteria is known to be sub-optimal within Gibbs. A similar grid search on step size and step number can be done for HMC-within-Gibbs for a fair comparison. 2. By inspecting the code, the MCMC component for discrete variables are found to be a MH step. This should be emphasised in the section. 3. Continuing from (2), different MCMC components for discrete variables should be experimented. For example, use the standard particle Gibbs (conditional-SMC) sampler as a drop-in replacement of the MH sampler. Although I still expect M-HMC would be better because continuous and discrete variables can be proposed jointly, which is not possible for HMC-within-Gibbs. ## How NUTS is used in 4.2 1. How adaptation for NUTS is set up? 2. The outperforming of M-HMC over NUTs may be due to NUTS stuck in some of the modes for too long. Have you inspected if that is the case? 3. Continuing from (2), if that the case, using a larger step size manually or lower the targeted acceptance ratio might be simply better (assuming currently dual averaging is used for adaptation). Finally, too many contents on the correctness of the proposed methods are in the appendix. It would be better to spend more content on a sketch of proof using the lemmas in the appendix.

Correctness: The proof of Theorem 1 (detailed balance) seems to be correct. The method looks empirically correct according to the experiments.

Clarity: Yes, both the method and experiments part of the main paper are easy to follow. While some details are missing in the experiment section, they can be found in the provided code. Besides, it would be nice if there is any illustration of the simulation of the trajectory of the proposed kernel itself. The main body of the proof lives in the appendix. In fact, a proper validity check of the algorithm would require going through the first 7-page of the appendix, which contains 4 lemmas, carefully. I think the paper should spent a good space on sketching the proof using lemmas and defer less contents on proof to the appendix. Maybe considering relating the lemmas in the paragraph beneath (1) is also a good idea? I would like to increase my score if a better sketch of the proof can be made available in the main paper.

Relation to Prior Work: Yes, the author(s) do(es) a good job on relating M-HMC to prior works in their method part and also compares most of the alternatives in the experiments.

Reproducibility: Yes

Additional Feedback: One suggestion of the presentation of Algorithm 1 is that it might be better to define the two integrators as separate algorithms, so that the main algorithm would look more modular and easy-to-follow. Also, where does the function Line 18 - 23 correspond to in the main paper or appendix? If it's less important which might be OK to move to the appendix? ######### # Update # ######### The author feedback looks good to me and I raised my score accordingly.


Review 3

Summary and Contributions: I have read the other reviews and author feedback. I decided to keep my score because more clear simulations or more theory would be helpful for making this proposed method trustful and easy to use. ----- This paper studies how to use Hamiltonian Monte Carlo (HMC) to sample mixed discrete and continuous distributions. It proposes M-HMC which creates one continuous variable for each binary discrete variable and then applies HMC on top of it. This paper also proves detailed balance property for M-HMC. It shows empirical evidence that M-HMC requires less sample than previously proposed DHMC.

Strengths: Proposed new M-HMC method to sample from mixed discrete and continuous distributions. Proved detailed balance property for M-HMC. Some empirical evidence for the fast convergence of M-HMC.

Weaknesses: 1. No theoretical insights about the speed-up M-HMC can achieve are provided. 2. It seems when the number of discrete variables becomes large, M-HMC will have to introduce much more continuous variables than DHMC. More continuous variables may make M-HMC slower in computation. Is it possible to provide a plot for both the actual computation time and sample size as a function of total number of discrete variables, for both DHMC and M-HMC? 3. Figure 3 shows that the performance of M-HMC is very dependent of the T choices. Any suggestions for the fix? 4. It is not clear at all whether the discrete part, after transforming it to be continuous variables, needs to use HMC. Neither is it clear why use HMC in the discrete part improves performance, because there is no gradient for the discrete part. Would a random walk type implementation enough for the discrete part?

Correctness: The proof of Theorem 1 looks correct to me.

Clarity: Yes. The paper writing is well structured. The traceplot in Figure 4 is not readable.

Relation to Prior Work: Yes.

Reproducibility: Yes

Additional Feedback:


Review 4

Summary and Contributions: This paper presents mixed Hamiltonian Monte Carlo (MHMC) for extending Hamilton Monte Carlo to models with discrete random variables. Unlike previous work on Discontinuous Hamiltonian Monte Carlo (i.e. DHMC, Nishimura et al 2019), MHMC does not embed discrete random variables into the reals because the DHMC embedding of non-ordinal discrete random variables tends to harm mixing rate (as previously noted by Nishimura 2019 section 2.2 and demonstrated empirically in sections 4.1/4.2). Instead, MHMC introduces auxiliary continuous random variables on the torus (viewed as a quotient of the hypercube) which function as "clocks" for performing a Metropolis-Hastings propose/accept of the associated discrete random variables whenever the auxiliary random variable hits the quotient boundary. By doing so, the authors claim that MHMC is able to update the discrete random variables "more frequently within HMC" (L125) and in a non-trivial way which does not "result in incorrect samples" (L126). Numerical experiments in section 4 are presented to substantiate these claims, demonstrating that (1) "naively" performing MH within HMC "doesn't work" (L186), (2) DHMC's mixing rate is sensitive to the ordering implied by the embedding, and (3) results in higher MRESS on a number of models considered (24d GMM, Bayesian logistic regression, correlated topic modeling). ===== POST REBUTTAL ====== It seems I misunderstood the notion of the clock. Line 112 says, "If qD hits either 0 or τ at site j (i.e. qjD ∈ {0,τ}), we propose a new x ̃ ∼ Qj(·|x), " This suggested a clock but on closer inspection of Line 10 in Algorithm 1, it appears that what the algorithm is doing is simply reordering the discrete random variables. Each discrete variable gets updated in each leapfrog step and the order of picking the discretes is different in each leapfrog iteration -- this is perfectly reasonable. So the comparison in Fig 1a and 1c is indeed correct and shows that a naive implementation of MH within HMC is not accurate which demonstrates value in the proposed MHMC scheme.

Strengths: The work is clearly a novel extension of previous work. The authors have demonstrated that we need to keep the discrete variables in the state unlike previous work where the auxiliary continuous variables encode the discrete variables. The empirical results are very reasonable as well.

Weaknesses: The distinction between this method and MH within HMC is not sufficiently strong for this work to be consider very novel. Essentially what the authors are proposing is an interleaving between HMC with a random choice being made to change the value of the discrete random variable (using a sampled kinetic energy and the number of steps taken to hit a fixed distance tau). If the choice is made to change the variable then the kinetic energy is also used to determine whether or not to actually change the variable based on the change in energy with the change in the discrete. So the only distinction between the above description and HMC interleaved with MH steps is that in each HMC step a new kinetic energy is sampled for the discrete variable and this is used to make the transition. One kinetic energy versus multiple seems quite simple and one must really consult the experiments to validate that this idea helps or not. Now the experiments support the idea but the implementation in Appendix L262 of the two versions don't quite match the description in the paper. For example where is the clock-like behavior that MHMC needs to select whether or not to make a transition. I could argue that the implementation in the Appendix L262 is not an ergodic MCMC. Ergodicity requires aperiodicity. A sufficient condition for aperiodicity is having a small self-transition probability. In most MCMC algorithms as long as self-transitions are possible we don't worry about periodicity but this implementation has me concerned. Adding the clock would have helped.

Correctness: The comparison of M-HMC to MH within HMC is questionable.

Clarity: The paper is very well written.

Relation to Prior Work: Comparison to prior work is well documented.

Reproducibility: Yes

Additional Feedback: Rather than comparing against MH within HMC, section 4 switches to NUTS within Gibbs (NwG). This introduces complications because any adaptation performed by NUTS may become inappropriate after discrete RVs are resampled. A more fair comparison would be against HMC within Gibbs (with the HMC step size and trajectory length optimized similarly to what is done for MHMC). Also, the provided code uses a deprecated `ElemwiseCategorical` API for the Gibbs step rather than `CategoricalGibbsMetropolis`. The authors criticize MH within HMC because "discrete variables can only be updated infrequently" (L30) to "suppress random walk behavior," MHMC equally suffers from this issue because if the discrete variables are frequently updated (e.g. if the kinetic energies of the associated auxilliary variables are large) then the trajectory length between successive discrete random variable updates is short.

[Author Response · NeurIPS 2020]

We thank the reviewers for their thoughtful feedbacks. We are encouraged they think M-HMC is addressing an important
problem (**R1**, **R2**), "is potentially going to be very influential" (**R1**) and is "a method with potential wide usage" (**R2**).
We are glad all reviewers find the paper to be well-written and clearly positioned w.r.t. prior work, the theory is correct
and "presented carefully" (**R1**), the experiments are "extensive" (**R1**), "well-designed" (**R2**), "very reasonable" (**R4**).
[**R2**] **...too many contents on the correctness of the proposed methods are in the appendix...I would like to**
**increase my score if a better sketch of the proof can be made in the main paper.** Great point. The main idea is
to decompose the transition probability kernel $R_T(s, B)$ into summation along countable number of deterministic
"probabilistic paths", then use similar proof techniques as in RRHMC. Starting from $s \in \Omega \times \Sigma$, for a given travel
time $T$, an M-HMC iteration (without MH correction) specifies a deterministic mapping for a fixed sequence of
random discrete proposals ($Y$ in appendix). We introduce "probabilistic path" ($\omega(s, T, Y)$ in appendix), containing all
information (sequence of proposals, indices/times&accept/reject decisions for discrete updates, and evolution of the
discrete/continuous states and auxiliary variables) of such deterministic trajectories for fixed $Y$'s. There can only be a
countable number of possible probabilistic paths, since $T$ and discrete state space are finite, so traveling from $s$ for time $T$
gives a countable number of possible destinations $s'$. This implies the decomposition $R_T(s, B) = \sum_{s'} \sum_Y r_{T,Y}(s, s')$,
where the summation is done over possible destinations $s'$ and all valid $Y$'s that bring $s$ to $s'$ through $\omega(s, T, Y)$.
$r_{T,Y}(s, s')$ is the transition probability along the deterministic trajectory $\omega(s, T, Y)$. Using similar proof techniques as
in RRHMC, we can prove detailed balance for these deterministic trajectories (Lemma 4 in appendix). This in turn
proves detailed balance of M-HMC. *We will expand and include the above proof sketch in camera-ready version.*
[**R4**] **The distinction between this method and MH within HMC is not sufficiently strong for this work to be con-**
**sidered very novel...the implementation in Appendix L262...don't quite match the description in the paper...The**
**comparison of M-HMC to MH within HMC is questionable.** This seems a *factual* misunderstanding. This compar-
ison is not meant to be novel. Rather, MH (with or without self-transition) within HMC isn't valid (Fig. 1c), while
M-HMC is (Fig. 1a). We can't imagine a stronger comparison. Appendix L262 is MH within HMC (L188, appendix
L178), so **R4** is correct, it's invalid. M-HMC is in appendix L256, which is valid and matches Algo. 1. To us, the
distinction being simple in this 1 discrete variable case is a strength. This simple change naturally comes out of Algo. 1,
and corrects the inherent bias in MH within HMC (Fig. 1a/1c). In fact, we specifically included this function to highlight
this simple distinction, showing M-HMC is correct, more efficient, yet extremely easy to incorporate into existing HMC
implementations. For novelty: **R1**, **R2** both think M-HMC solves an important problem and can potentially be very
influential and see wide usage. Even **R4** mentioned "The work is clearly a novel extension of previous work".
[**R4**] **M-HMC equally suffers from this issue (long trajectories for random-walk suppression vs frequent discrete**
**updates) because if the discrete variables are frequently updated...then the trajectory length between successive**
**discrete random variable updates is short.** We respectively disagree. In HMC, using long trajectories gives distant
proposals/random-walk suppression (due to consistent momentum within an iteration), but means infrequent discrete
updates. Shorter trajectories enable more frequent discrete updates, but frequent momentum resampling between
iterations increases random-walk behavior. In contrast, M-HMC can always utilize consistent momentum/kinetic energy
in long trajectories, with potentially frequent discrete updates, to get distant proposals/random-walk suppression. The
trajectory lengths between successive discrete updates are irrelevant, as these updates involve no momentum resampling.
[**R3**] **No theoretical insights about the speed-up...** Intuitively speed-up is from more frequent discrete updates. More
detailed theoretical analysis is good future work. [**R3**] **...M-HMC will have to introduce much more continuous**
**variables than DHMC...** Using Algo. 1, we only need 1 continuous variable ($k_i^{\mathcal{D}}$ in Algo. 1) for each discrete
variable. DHMC needs 2 ($q_i^{\mathcal{D}}$ in L85, and $p_i^{\mathcal{D}}$). M-HMC also doesn't need to update all discrete variables every time,
making it more efficient than DHMC (see L265-268 for comparison on BLR). [**R3**] **...performance...very dependent**
**of...$T$...suggestions...?** HMC is known to be sensitive to $T$ (and step size). So is M-HMC. Automatically picking
these parameters is important future work (L325). [**R3**] **It's not clear at all...the discrete part...needs...HMC...** This
mechanism enables discrete updates within HMC, which, if done naively, is invalid. The proposals $Q_i$'s are used instead
of gradients information. A random walk type implementation would be the invalid MH within HMC (Fig. 1c).
[**R2**, **R4**] **HMC-within-Gibbs (HwG) should be used instead of NwG as baseline** Additional experiments with HwG
show M-HMC is 2.5 (24D GMM)/3.6 (BLR)/3.3 (CTM) times more efficient than HwG in MRESS. HwG is indeed
better than NwG as **R2** suggested. *We will include HwG as an additional baseline in camera-ready version.* [**R2**]
**...MCMC component for discrete...should be emphasised...** Will do. [**R2**] **...different MCMC components for**
**discrete variables should be experimented.** We have started (and will include in camera-ready version) experiments
with *Turing.jl* for particle Gibbs. But we share **R2**'s expectation that M-HMC would still be better. [**R2**] **How**
**adaptation for NUTS is set up?** Dual averaging, 0.8 target acceptance. [**R2**] **The outperforming of M-HMC over**
**NUTS may be due to NUTS stuck in some of the modes...using a larger step size manually...might be simply**
**better...** This turns out to be exactly the case. NUTS MRESS increased to $1.37 \times 10^{-3}$ with step size 5 (v.s. 4 from
dual averaging), outperforming M-HMC as one expects. *We will update the paper with these observations.*
[**R1**] We will add a concise summary of experiments. [**R2**] We visualized the kernel for 1d GMM in 4.1, showing
discrete updates and distant proposals. We will include the visualization, along with suggested proof/algorithm
re-organizations. [**R3**] We will keep only Fig. 4a for 1 chain in main paper to improve readability of the traceplots.

[Meta-Review · NeurIPS 2020]

This paper concerns using HMC on distributions with mixed discrete and continuous variables. Previous papers usually either (i) alternate continuous and discrete updates or (ii) relax discrete variables into continuous. This paper argues that both of these approaches tend to lead to slow mixing. Instead it proposes to augment the discrete variables with continuous variables with a torus topology, the position of which determines when discrete updates can be performed (maintaining the Hamiltonian). This is proven to maintain detailed balance. Intuitive arguments are given for faster mixing, along with some numerical evidence. There is no theoretical proof of faster mixing. (To be sure, mixing time proofs are very challenging!) The reviewers had a consensus in favor of accepting the paper. The AC would like to echo a couple points from the reviews, in the hope that the final impact of the paper can be maximized. First, the paper is challenging to read -- the AC was able to understand the big picture only after reading the reviews. Second, the paper could more strongly emphasize intuition for why mixing times would be faster for this work. Finally, it would be great if the experimental results could convey a bit more "insight" into "why" the new algorithm mixes faster.